

# Implementing a process-based representation of soil water movement in a second-generation dynamic vegetation model: application to dryland ecosystems (LPJ-GUESS-RE v1.0)

Wim Verbruggen[1], David Wårlind[2], Stéphanie Horion[1], Félicien Meunier[3,4], Hans Verbeeck[3], and Guy Schurgers[1]

[1] Department of Geosciences and Natural Resource Management, University Of Copenhagen, Denmark
[2] Department of Physical Geography and Ecosystem Science, Lund University, Sweden
[3] Q-ForestLab, Department of Environment, Ghent University, Ghent, Belgium
[4] Isotope Bioscience Laboratory, Ghent University, Ghent, Belgium

*Correspondence to*: Wim Verbruggen (verbruggen.wim@gmail.com)

**Abstract.** Dryland ecosystems are globally important, yet state-of-the-art dynamic vegetation models often lack specific processess or parameterizations that are critical for accurately simulating dryland dynamics. These missing processes include a realistic calculation of soil water movement, detailed plant-water relations, or a representation of deep water uptake. In this study we show how including a process-based soil hydrology scheme in the LPJ-GUESS (Lund-Potsdam-Jena General Ecosystem Simulator) model can improve its usefulness for simulating the functioning of dryland ecosystems. By replacing the default 15-layer bucket representation of soil hydrology in LPJ-GUESS v4.1 with a mechanistic description of soil water movement based on the 1D Richards Equation, we show that the model is better able to capture seasonal patterns of water cycling through dryland ecosystems at both the site level and the regional level. In addition, the inclusion of a new set of bottom boundary conditions, such as a permanent groundwater layer, further expands the range of ecosystems the LPJ-GUESS model can simulate. We show that soil bottom boundary conditions, in particular varying levels of groundwater depth, can have a large influence on vegetation composition and water cycling. Our new model developments open new avenues to simulate dryland ecohydrology more realistically.

## 1 Introduction

Dryland ecosystems are globally important, as they account for about 40% of Earth's terrestrial surface and net carbon uptake, while sheltering more than 30% of the human population (Gilbert, 2011; Grace et al., 2006; Wang et al., 2012). Drylands have been shown to drive the interannual variability and long-term trend of the global land carbon sink (Ahlstrom et al., 2015; Poulter et al., 2014) and a 10% increase in vegetation cover in semi-arid lands has been observed globally over the past decades (Ruehr et al., 2023). Recently it has been shown that more than 30% of global dryland ecosystems are dependent on access to groundwater, including several important global biodiversity hotspots, while more than half of these groundwater-dependent ecosystems are located in regions with declining groundwater trends (Rohde et al., 2024). Unsurprisingly, drylands are known



to support trees with the deepest root systems over all biomes globally, with observed rooting depths down to 60 m in the soil, providing access to the groundwater table (Do et al., 2008; Fan et al., 2017).

Dynamic vegetation models (DVMs) are process-based tools that can help to gain deeper insights into the functioning of dryland ecosystems and their link with soil hydrology. By integrating a multitude of processes from the leaf level (e.g.,
photosynthesis) up to the ecosystem level (e.g. competition, carbon cyling) these models help to quantify the role of various biomes in the global carbon and water cycle, study vegetation demographic changes, as well as predict ecosystem response to future climate scenarios (Prentice et al., 2007). Several studies used DVMs to study dryland ecosystems, either as their main biome of focus or within the context of global studies (Ahlstrom et al., 2015; Baudena et al., 2015; Boke-Olén et al., 2018; Brandt et al., 2017, 2018; Dashti et al., 2021; Haverd et al., 2017; Hickler et al., 2005; Lehsten et al., 2016; Meunier et al.,
2022; Scheiter et al., 2019; Seaquist et al., 2009; Verbruggen et al., 2021a, b, 2024). However, only a few studies updated the parameterization and evaluated the performance of DVMs for drylands specifically (Dashti et al., 2021; Verbruggen et al., 2021a, b). Furthermore, only limited attention has been given to identifying the important processes for reliably simulating dryland ecohydrology (Whitley et al., 2017).

In this study we focus on the Lund-Potsdam-Jena General Ecosystem Simulator (LPJ-GUESS) version 4.1 dynamic vegetation
model (Smith et al., 2001, 2014). This model was used in several of earlier mentioned the dryland studies, where it was shown to be capable of simulating tree-grass co-existence in savannas, to reasonably simulate carbon and water fluxes at the site- and regional level, and to capture the overall greening trends in the Sahel (Hickler et al., 2005; Verbruggen et al., 2021a, 2024). The model's hydrological representations have also been evaluated favourably against global data products of runoff, evapotranspiration and near-surface soil moisture (Gerten et al., 2004; Zhou et al., 2024).

Despite its good overall performance, a few important processes are still missing in LPJ-GUESS to capture dryland ecohydrology reliably. These developments are needed to make the model more useful and realistic for future projections of drylands under changing climatic conditions, as well as correctly capturing the competition for soil water of different vegetation types. A first fundamental limitation of the model is its oversimplied representation of soil hydraulics, i.e. the dynamics of soil water through the different layers. Plant water uptake from the soil through their roots is a critical process for
water-limited ecosystems such as drylands. Therefore, if the basic physics of soil water dynamics are poorly represented in an ecosystem model, the model will struggle to correctly capture and project the vegetation response to changes in soil water conditions, such as drought or high rainfall extremes. While the current version (v4.1) of LPJ-GUESS already improved the resolution of the soil layers from 2 coarse layers of 0.5 m and 1.0 m thickness (v4.0) (Gerten et al., 2004) to 15 layers of 0.1 m thickness (Zhou et al., 2024), and while the global model output is evaluated favourably against observations, the dynamics
of water percolation between the simulated soil layers are based on a bucket model. However, most of the DVMs today represent soil water movement based on gradients of soil water potential (Richards equation).

In this paper we show that the bucket model simplification in LPJ-GUESS creates unrealistic soil water dynamics for drylands, as well model artifacts, such as a discontinuous average soil water profile that enables the model to simulate tree–grass coexistence for the wrong reasons. We solve these issues by implementing a new soil hydrology scheme for the LPJ-GUESS



model. This new soil hydrology simulates mass-conservative movement of soil water based on gradients in water potential by adopting the implementation of Ireson et al. (2023) to solve Richards equation in LPJ-GUESS. We keep the number of soil layers fixed to 15, but the thickness of the different layers can now be changed, opening up the model to simulate different soil depths. Our new model version also allows the simulation of two additional bottom boundary conditions, besides the default free drainage condition: bedrock and aquifer. The bedrock condition does not allow for any water to percolate out of the system

by baseflow runoff, while the aquifer condition simulates an additional layer of groundwater beneath the bottom layer. These improvements allow the model to simulate a variety of drylands conditions, ranging from shallow soils to deep groundwater-dependent ecosystems.

After introducing these new model developments, we evaluate the new model against observations of dryland carbon and water cycles. In particular, we compare the model outputs with site-level fluxtower data from Senegal as well as global data

products, focusing on the Sudan-Sahel region. Finally, we perform a few sensitivity tests on the new model. In earlier work we have shown that terrestrial biosphere models have only low sensitivity to soil texture in the tropics (Meunier et al., 2022) and in this current paper we test whether changing the soil hydrology has any impact on this sensitivity. For a second sensitivity test we investigated how changing groundwater table depths (GWTD) may influence simulated vegetation cover and surface hydrology. By doing this we show how our new hydrology scheme opens up the model capability for simulating soil water

dynamics in groundwater-dependent dryland ecosystems.

## 2 Methods

### 2.1 Focus area: Sudan-Sahel region and the Dahra fluxtower site

The Sudan-Sahel region is an ecoclimatic transition zone located between the Sahara Desert and the humid Guinean zone (**Figure 1**). The northern Sahel region is defined by the 150 and 600 mm mean annual precipitation (MAP) levels as its northern

and southern boundaries. For the southern Sudanian zone the annual rainfall varies between 600 and 1000 mm on average (Karlson and Ostwald, 2016) (**Figure 1**). The vegetation cover follows this strong North-South precipitation gradient, varying from grassy savannas and shrublands in the north to open dry forests in the south (Souverijns et al., 2020). Most rainfall occurrence is limited to a short wet season, which usually takes place between June and October.

For the site-scale simulations and model evaluation we focused on the Dahra fluxtower site, located in the Sahelian zone of

Senegal (15°24′10″ N, 15°25′56″ W). This site is a grazed semiarid savanna with a mean annual rainfall of 416 mm which mainly occurs during a short rainy season (July–September). It is equipped with a study tower measuring meteorological, hydrological and radiation sensors, as well as an eddy covariance system for continuously measuring carbon, water and energy fluxes since 2010 (Tagesson et al., 2015; Wieckowski et al., 2024).



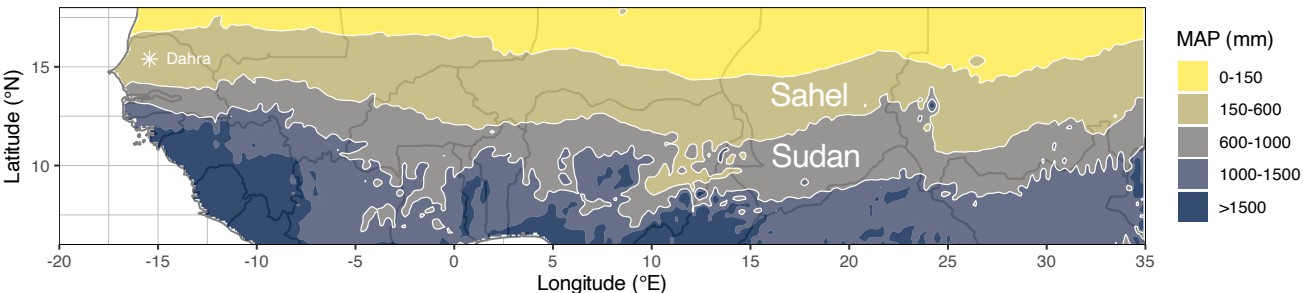

**Figure 1.** Map of the Sudan-Sahel region in West and Central Africa, including the mean annual precipitation levels that define the Sahelian (150–600 mm) and the Sudanian (600–1,000 mm) regions. Rainfall was obtained from ERA5-Land data (Muñoz-Sabater et al., 2021) and averaged over the entire timespan (1950–2022). Location of the Dahra, Senegal eddy covariance fluxtower (15°24′10″ N, 15°25′56″ W) is marked by an asterisk.

## 2.2 Baseline model

### 2.2.1 LPJ-GUESS v4.1

The Lund-Potsdam-Jena General Ecosystem Simulator (LPJ-GUESS) v4.1 is a dynamic vegetation model which simulates global vegetation cover and functioning, together with its associated water, carbon and nitrogen cycles (Smith et al., 2001, 2014). Global vegetation is represented by a set of plant functional types (PFTs) which group all terrestrial vegetation species in broad classes of functional similarity. Heterogeneity in size-age classes within each PFT is accounted for by using cohorts as the basic vegetation unit in the model. Vegetation growth is primarily driven by photosynthesis, and modulated by competition for light, soil water and soil nitrogen between cohorts. The smallest unit of explicit spatial information is the gridcell, whose size depends on the spatial resolution of the meteorological and soil input data (here 0.1° × 0.1°). Within each gridcell, the model simulates a large number (here 100) of replicate patches (size 1000 m²) in order to average out the impact of patch-level stochastic disturbance events, which will create different life histories between patches. Plant ecophysiological processes and soil hydrology are resolved at a daily time step, while carbon allocation and vegetation dynamics are accounted for at the end of each simulated year.

The model couples the calculation of leaf-level photosynthesis with stomatal conductance. Plant water status is represented as the ratio between root-zone water supply and canopy water demand. Low values of this ratio will cause the vegetation to be drought stressed, leading to a reduction of photosynthesis by stomatal closure. This ratio will also modulate the carbon allocation between the above- and belowground plant tissues, as well as trigger leaf growth and abscission for raingreen PFTs. The PFT parameters of the LPJ-GUESS model have previously been updated for the Sudan-Sahel region, both across a network of site-level fluxtowers (Verbruggen et al., 2021a) and at the regional scale (Verbruggen et al., 2021b, 2024). For this study we again use this updated regional parameter set, simulating the following PFTs: $C_4$ grass, Tropical Evergreen Trees, Tropical Raingreen Trees, and Tropical Shrubs (Supplementary Table 1).



### 2.2.2 Standard soil hydrology

Soil hydrology in the default version of LPJ-GUESS v4.1 is represented by either a two-layer model or a multi-layer model. The original two-layer scheme simulated two soil layers of thickness 50 cm and 100 cm respectively (Gerten et al., 2004) while the multi-layer scheme provides a greater vertical resolution by simulating 15 layers of 10 cm thickness each (Zhou et al., 2024). The soil water available for plants in each layer is modelled as the fraction of water content ($0 \leq wcont \leq 1$) between the wilting point ($\theta_{WP}$) and field capacity ($\theta_{FC}$):

$$wcont = \frac{\theta - \theta_{WP}}{\theta_{FC} - \theta_{WP}} \qquad (1)$$

where $\theta$ is the soil water content (m³/m³) in each layer, and is limited to [$\theta_{WP}$, $\theta_{FC}$] (see also Supplementary Materials S1.1). For each patch and each daily timestep, plants can transpire water from all layers, depending on soil water content and PFT root fraction for each soil layer, as well as patch-level water demand, cohort water stress status, cohort foliar projective cover (FPC), and a PFT parameter *emax* (mm/d) which will limit daily transpiration (Supplementary Materials S1.2) (Sitch et al., 2003). Vertical root distribution is modelled by an asymptotic equation ($RD_{cumul} = 1 - \beta_{root}^{z}$) which calculates the cumulative root fraction ($RD_{cumul}$) downward in function of depth ($z$) and a PFT-specific shape parameter $\beta_{root}$ (Jackson et al., 1996). If this cumulative fraction does not reach unity at the bottom layer, the missing root biomass fraction is assigned to the bottom layer. Water can evaporate from the top two soil layers (20 cm) depending on remaining water content, atmospheric water demand (daily equilibrium evaporation) and the bare soil fraction of the patch (Gerten et al., 2004; Rost et al., 2008). See supplementary materials S1.3 for more details.

Rainfall adds water to the system, part of which will be intercepted by the vegetation leaf cover, from which it evaporates. Remaining rain water and snow melt reaches the upper soil layers (0–50 cm), which are replenished according to their water holding capacity. Excess water above field capacity is removed from the system as surface runoff. Percolation transports water downward to deeper soil layers and is calculated as a power law in function of available water content:

$$perc(l) = p_b * wcont(l)^{p_e} \qquad (2)$$

where $p_b$ and $p_e = 2$ are shape parameters (see Sup. Mat. S1.1). For both model configurations, water percolates from the upper soil layers (0–50 cm) to the lower layers (50–150 cm). For the multi-layer representation, the percolated water is distributed between the sublayers, according to their water holding capacity. Notably, percolation is limited to days for which the sum of rainfall and snow melt is higher than 0.1 mm (hard-coded) (Nord et al., 2021). A fraction of plant-available water in the lower layers (50–150cm) further percolates out of the system as baseflow runoff. Finally, excess water in each layer dissipates out of the system as lateral flow runoff.



Calculated soil hydraulic properties include the water content at wilting point and field capacity, as well as the parameters of
the percolation power law. These parameters are calculated by pedotransfer functions from soil textural data, which are given
as an input for the model and are assumed to be constant over all soil layers (Cosby et al., 1984; Gerten et al., 2004), see
Supplementary Materials S1.1.

**2.3 Soil hydrology model updates**

**2.3.1 Soil layer structure and water representation**

By default, our updated version of the model, presented in this paper, still uses 15 soil layers of 10 cm thickness. However,
the number and thickness of soil layers can now be adjusted individually without affecting the simulated hydrological
processes, i.e. all layers – except the top and bottom layers – are processed equally and no layers are grouped together. Soil
water content $\theta$ can vary between the water content at wilting point ($\theta_{WP}$) and saturation ($\theta_S$). The model simultaneously tracks
$\theta$, the soil water potential $\psi$ ($m$) and the hydraulic conductivity $k$ (m s$^{-1}$) for each layer, based on the Campbell (1974) relations:

$$\psi = \psi_s(\theta/\theta_s)^{-b} \tag{3}$$
$$K = K_s(\theta/\theta_s)^{2b+3} \tag{4}$$

where $\psi_S$, $\theta_S$ and $K_S$ are parameters which represent the soil water potential ($m$), content (m$^3$ m$^{-3}$) and soil hydraulic conductivity
(m s$^{-1}$) at saturation, respectively. $b$ is an empirical parameter (Campbell, 1974). These four parameters are derived from soil
texture using pedotransfer functions from Cosby et al., (1984) and Romano and Santini (2002) (Supplementary Materials
S1.1). To retain compatibility with processes outside soil hydrology (e.g. plant water uptake), the water content fraction *wcont*
is still calculated from $\theta$ using Eq. (1). Soil textural data and the derived soil hydraulic parameters are assumed to be constant
over all layers, but this can be changed by making a few modifications to the model code.

**2.3.2 Soil water dynamics based on Richards equation**

Soil water movement between layers is based on gradients in soil water potential, together with a gravitational term. It is
calculated by integrating the change in $\psi$ over every daily model timestep, using a $\psi$-based form of the 1D Richards equation:

$$\frac{\partial\psi}{\partial t} = \frac{1}{C(\psi)}\frac{\partial}{\partial z}\left(K(\psi)\left(\frac{\partial\psi}{\partial z} - 1\right) - S\right) \tag{5}$$

where the specific moisture capacity $C(\psi) = \mathrm{d}\theta/\mathrm{d}\psi$ and hydraulic conductivity $K(\psi)$ are calculated from the Campbell relations
above (Eqs. (3) and (4)) (Celia et al., 1990; Ireson et al., 2023) and the sink term $S$ represents water uptake by plant roots in
our model, although this term can include water release as well (e.g. to account for hydraulic lift) in a future version of the



model. For the discrete solution of this ordinary differential equation (ODE) we based our implementation on the mass-conservative "openRE" approach, using a sub-daily adaptive timestep ODE solver (Ireson et al., 2023). The following workflow is modified from Ireson et al. (2023) in order to account for variable layer thickness (Δz; units: *m*).

We assume a system of N soil layers, with an index *i* that varies between the surface (*i=1*) and the bottom layer (*i=N*). The rate of change in soil water potential $\psi_i$ (*m*) for each soil layer *i* is given by:

$$\left.\frac{\partial \psi}{\partial t}\right|_i = -\frac{1}{C(\psi_i)}\left.\frac{\partial q}{\partial z}\right|_i = \frac{1}{C(\psi_i)}\frac{q_{i-1,i} - q_{i,i+1} - Et_i}{\Delta z_i} \tag{6}$$

where $\partial q$ is approximated by the balance of incoming ($q_{i-1,i}$) and outgoing ($q_{i,i+1}$) water fluxes. $Et_i$ is the implementation of the sink term (*S*) in Eq. (5), here accounting for plant transpiration from layer *i* (m day$^{-1}$), and $\Delta z_i$ is the thickness (m) of layer *i*. For the internal layers (*i*∈[2,N-1]) the incoming and outgoing fluxes have a similar functional form, as the flux from layer *j* to layer *k* is given by:

$$q_{j,k} = -\frac{K(\psi_k) + K(\psi_j)}{2}\left(\frac{\psi_k - \psi_j}{\Delta z_{j,k}} - 1\right) \tag{7}$$

where $\Delta z_{j,k} = (\Delta z_j + \Delta z_k)/2$ is the center-to-center layer distance between both layers. Note that these fluxes can go in any direction, so the "incoming" flux may as well be an outgoing water flux from layer *k* to *j* if the water potential gradient is strong enough. As discussed by Ireson et al. (2023), we use the arithmetic mean of *K* at the layer center points, but other formulations, such as the harmonic mean, are possible as well (Ireson et al., 2023). The fluxes at the top and bottom layers (*i*∈{1,N}) are determined by the boundary conditions.

### 2.3.3 Boundary conditions

The incoming water flux for the top layer ($q_1$) consists of the incoming net infiltrating water and snowmelt from the surface ($W_{in}$), minus the outgoing soil surface evaporation ($Es$) from this first layer, both with units [m day$^{-1}$]

$$q_1 = W_{in} - Es \tag{8}$$

For the bottom boundary condition ($q_N$) we provide three possible options: free-drainage, bedrock, or aquifer.

Under the *free-drainage* condition we assume that the layers below the bottom layer are in hydrological equilibrium with the bottom layer, i.e. they have the same hydraulic conductance $K_N$ and water potential $\psi_N$ as the bottom layer. Hence, there is no water potential gradient and water percolation is therefore solely driven by gravity. Setting $\psi_j = \psi_k = \psi_N$ in Eq. (7), we obtain:



$$q_N = K(\psi_N) \tag{9}$$

For the *bedrock* boundary condition, we model an artificial bedrock layer below the bottom layer through which no water can be tranported. This is implemented by the condition that the flux from the bottom layer to the layers below is zero:

$$q_N = 0 \tag{10}$$

Finally, for the *aquifer* bottom boundary, we assume that the layers below the bottom layer are fully saturated, ie. $\psi_k = \psi_s$ in Eq. (7), which then reduces to

$$q_N = -\frac{K_s + K(\psi_N)}{2}\left(\frac{\psi_s - \psi_N}{\Delta z_N/2} - 1\right) \tag{11}$$

where $K_s = K(\psi_s)$ is again the hydraulic conductivity at saturation, and $\Delta z_N/2$ is the distance between the bottom layer center and the aquifer. This boundary condition can act as an additional source of soil water, as water can be transported upward into drier soil layers above whenever a strong gradient in in soil water potential emerges.

## 2.3.4 Evaporation and runoff

The calculation of surface evaporation ($Es$), transpiration ($Et = \sum_i Et_i$) and interception loss ($Ei$) is unaltered from the original model version, with the difference that surface evaporation only occurs from the top layer (10 cm) (Supplementary Materials S1). The sum of these components is the total evaporation ($E=Es+Et+Ei$) (see also Miralles et al. (2020) for a discussion on this terminology). The removal of water by $Es$ and $Et$ is implemented inside the ODE solver routine, as described earlier. Precipitation ($P$) that is not intercepted reaches the top soil layer and replenishes the water content of this layer until it reaches saturation ($\theta_{sat}$). Any excess above $\theta_{sat}$ is removed as surface runoff ($R_{surf}$), so the net water infiltration ($W_{in}$) is given by:

$$W_{in} = P - Ei - R_{surf} \tag{12}$$

At the end of each simulated day, a fraction ($f_{drain}$) of excess water content above field capacity in each layer $i$ is removed as lateral drainage ($R_{drain,i}$), following a similar implementation in the Community Land Model version 5 (Lawrence et al., 2019). This fraction is calculated as the tangent of the terrain slope, multiplied by a lateral flow parameter (default value 1). Terrain slope is set to a fixed value 2° but the code can be easily adapted to read this value from a map for each gridcell. The drainage fraction therefore has a default value of $f_{drain} = 0.034$ in the current implementation, so by default 3.4% of the excess water is removed as lateral drainage. Together with the baseflow runoff from the bottom layer ($R_{base} = q_N$) these three components form the total runoff ($R = R_{surf} + R_{drain} + R_{base}$, where $R_{drain} = \sum_i R_{drain,i}$).





### 2.3.5 Numerical integration and water mass balance

The LPJ-GUESS v4.1 model runs with a daily timestep, during which hydrological processes – such as surface evaporation, plant transpiration and runoff – as well as several vegetation-related processes are calculated. Between these daily timesteps,

our updated soil hydrology scheme calculates the percolation between layers using the framework described above, using a an ODE solver with an adaptive sub-daily timestep in order to minimize integration errors (see further) (Ireson et al., 2023). During each timestep of the LPJ-GUESS model, the daily hydrological processes used in the equations above ($W_{in}$, $E_t$, $E_s$) are passed on as constants to the sub-daily RE integrator, which automatically converts them from daily to subdaily rates, depending on the number of sub-daily integration timesteps (NTS) the ODE solver uses (i.e. multiplication by dt = 1/NTS).

At every daily timestep, the water mass balance error ($\varepsilon_{WB}$) for the entire soil column is calculated as the difference between the different incoming and outgoing water fluxes and the total soil column water storage $\Delta\theta$ term:

$$\epsilon_{WP} = P - E - R - \Delta\theta \tag{13}$$

where the storage term $\Delta\theta$ represents the change in soil water content compared to the previous daily timestep, and this will be the main contributor to $\varepsilon_{WB}$ due to numerical integration errors in the solution of Richard's equation.

To ensure water mass balance closure we follow the procedure from Ireson et al. (2023). We include the cumulative boundary fluxes ($Q_I$ and $Q_N$) to our system of ODEs, which are calculated as the sum of $q_I$ and $q_N$ over all daily time steps since the start of the simulation:


$$Q_j = \sum_{t=0}^{t_{now}} q_j(t) \tag{14}$$

for j∈{1,N}. Therefore, the complete system of ODEs that is integrated over any given daily time-step $t$ is given by:


$$
\begin{pmatrix}
\dfrac{dQ_1}{dt} \\[6pt]
\dfrac{d\psi_1}{dt} \\[6pt]
\dfrac{d\psi_2}{dt} \\[6pt]
\vdots \\[6pt]
\dfrac{d\psi_{N-1}}{dt} \\[6pt]
\dfrac{d\psi_N}{dt} \\[6pt]
\dfrac{dQ_N}{dt}
\end{pmatrix}
=
\begin{pmatrix}
q_1 \\[6pt]
-\dfrac{1}{C(\psi_1)}\dfrac{\partial q}{\partial z}\Big|_1 \\[6pt]
-\dfrac{1}{C(\psi_2)}\dfrac{\partial q}{\partial z}\Big|_2 \\[6pt]
\vdots \\[6pt]
-\dfrac{1}{C(\psi_{N-1})}\dfrac{\partial q}{\partial z}\Big|_{N-1} \\[6pt]
-\dfrac{1}{C(\psi_N)}\dfrac{\partial q}{\partial z}\Big|_N \\[6pt]
q_N
\end{pmatrix}
\tag{15}
$$



The ODE integrator solves this system for the water potentials $\psi_i$ in each soil layer $i$, as well as the cumulative boundary fluxes $Q_1$ and $Q_N$. To integrate this system of ODEs we used the *runge_kutta_cash_karp54* adaptive timestepper from the *odeint* library in the *boost* C++ package (Ahnert and Mulansky, 2011). A code snippet with an overview of the basic implementation

is given in Supplementary Materials S2, and the full model code can be accessed from a Zenodo archive (Verbruggen et al., 2025).

### 2.4 Model setup and forcing data

To assess the impact of these changes in the hydrological scheme on dryland ecosystem dynamics, model simulations were performed at two distinct spatial scales. Site-scale simulations were run for the Dahra fluxtower site in Senegal, while we also

performed regional simulations over the entire Sudan-Sahel region (**Figure 1**). The parameterization of the plant functional types in LPJ-GUESS has previously been optimized for both the site level (Dahra, Senegal) and the Sudan-Sahel region, and we continued to use these parameters for both the site-level and regional simulations (Verbruggen et al., 2021a, 2024). An overview of these parameters is given in Supplementary Materials S3.

The LPJ-GUESS v4.1 model is driven by daily averages of air temperature (°C) at 2 m height, incoming short-wave radiation

(W m$^{-2}$) and precipitation rate (mm day$^{-1}$). Site-level simulations used the meteorological data from the Dahra fluxtower site in Senegal, which were measured at 30 minute intervals for the the 2002–2022 period (Tagesson et al., 2015). We averaged these measurements over each day and used a gap-filling procedure to obtain a continuous driver data set for the entire period 2002–2022 (Supplementary Materials S3). Soil texture input (95.04% sand, 4.61% silt, 0.35% clay) was obtained from local measurements (Tagesson et al., 2015).

Regional scale simulations were driven by daily averaged ERA5-Land meteorological data for the 1950–2022 period at a 0.1° spatial resolution (Muñoz-Sabater et al., 2021). Regional soil texture data were obtained from the 250 m resolution ISRIC Africa SoilGrids database for six soil depths (Hengl et al., 2015). We averaged these soil texture data over all depths and regridded the data to match the meteorological driver grid.

For both spatial scales, we started the model simulations from bare soil with a 500-year spinup phase, at which the atmospheric

CO$_2$ level was fixed at 296 ppm, corresponding to the 1901 level. For the site scale we used the first 10 years of the Dahra fluxtower meteorological driver time-series in a cycle for the spinup, while for the regional scale we used the first 30 years of the ERA5-Land data product. Air temperature was detrended for both spinup drivers. This spinup phase was followed by a historical simulation (1901–2022) using the historical increase in the atmospheric CO$_2$ level (Friedlingstein et al., 2023) but still using the shortened meteorological drivers from the spinup-phase. Once the starting date of the original meteo data is

reached (i.e. the year 2002 for the Dahra drivers, and 1951 for the ERA5-Land data product) the full meteorological time-series are used to drive the model simulations until the year 2022.

All simulations were performed for the model versions with the original ("Default") and the updated ("RE") soil hydrology schemes, in order to enable comparison of both model versions and to analyse the impacts of the new model developments on



dryland ecohydrology. For evaluating the models against observation-based data products (see next section), we only used the
"free drainage" bottom boundary condition for the RE-based model, as this condition is the closest to the one used by the
original model.

## 2.5 Evaluation data

To evaluate the two model versions at the site level, we used water and carbon fluxes, as well as soil moisture data from the
Dahra fluxtower site, which were measured simultaneously with the meteorological drivers using the eddy covariance
technique and soil moisture sensors (Tagesson et al., 2015; Wieckowski et al., 2024). From these we also calculated and
evaluated the water use efficiency (WUE), defined as the ratio of daily GPP to ET ($WUE = GPP/ET$) after filtering out days
with an ET below 0.01 mm. At the regional scale we evaluated the model against the GLEAM v3.8a data product over the
period 1980–2022 (Martens et al., 2017; Miralles et al., 2011). We averaged model output (originally at 0.1° resolution) to
match the 0.25° grid used by GLEAM and compared total evaporation (E), soil evaporation (Es), plant transpiration (Et). The
GLEAM data product also includes assimilated surface soil moisture from the Climate Change Initiative (CCI) programme of
the European Space Agency (ESA) (Dorigo et al., 2017; Gruber et al., 2017), against which we evaluated our soil moisture
simulations. We also compared total vegetation leaf area index (LAI) in function of MAP against remotely sensed
measurements from the Moderate Resolution Imaging Spectroradiometer (MODIS) (Myneni et al., 2021) which were obtained
using the "appeears" software package (Hufkens, 2023).

## 2.6 Model sensitivity tests

### 2.6.1 Soil texture

Soil hydraulic properties in the LPJ-GUESS v4.1 model are derived from soil texture using pedotransfer functions (Cosby et
al., 1984; Smith et al., 2014). These include soil water content at wilting point ($\theta_{WP}$), field capacity ($\theta_{FC}$), saturation ($\theta_{SAT}$),
hydraulic conductivity at saturation ($K_S$), and the slope ($b$) of the soil water retention curve (Supplementary Materials S1). We
tested the model sensitivity to soil texture for both soil hydrology representations. To do so, we made a series of simulations
for the Dahra fluxtower site in Senegal, where we replaced the actual soil texture by all possible combinations of sand/clay/silt
contents, and analyzed the simulated ecosystem response. Note that resulting soil hydraulic properties are the same between
both model versions: the only difference is due to the different soil hydrology processes. For the RE-based model we used the
"free drainage" bottom boundary condition, which matches the bottom boundary condition of the default model version the
most closely. For the response variables we analyzed vegetation leaf cover (LAI) and the different evaporation components.
We compared the sensitivity of the default model version with our RE-based update by calculating the mean, standard deviation
and coefficient of variation of the model outputs over all soil textures, based on the averaged values over all years. Results
were visualised using ternary plots, including a scaling by the maximum (over all soil textures) value of model output, in order
to facilitate comparison of the sensitivities of the different analyzed output variables to soil texture.




### 2.6.2 Groundwater table depth

The new model version allows for using different bottom boundary conditions at any soil depth. For a second sensitivity test we analyzed the impact of GWTD on dryland vegetation by activating the aquifer bottom boundary condition and running the model for different soil depths. Soil depth was varied by keeping the number of layers constant to 15 but changing the thickness of the bottom 10 layers. We again used the site simulation for Dahra as baseline, but now imposing an aquifer at depths ranging from 0.75 m to 6 m in steps of 0.25 m, and analyzed the simulated vegetation cover (LAI) and evaporation components, as well as the different runoff components in function of water table depth. We also analyzed the influence of groundwater depth on soil moisture and root water uptake for each soil layer, separated into the dry and wet season.

## 3 Results

### 3.1 Model performance at Dahra fluxtower site

A comparison of the average yearly cycles of the evaporation and runoff components revealed significant differences between both model versions (**Figure 2**). The sum of the evaporation total (E) over the year was similar (around 300 mm/y) for both model versions, comprising a significant portion of average yearly rainfall (416 mm). While the updated model version had a higher E (values up to 3.9 mm/day) during the rainy season, compared to the default model (3.5 mm/day), this increase was compensated by a shorter tail in E after the rainy season (**Figure 2**b). The higher wet-season E in the RE model was caused by a higher soil surface evaporation rate (Es), while the longer subsequent tail in the default model was caused by dry-season transpiration (Et) by tree PFTs from deeper soil layers. Transpiration during the wet season was also overall lower (by 0.15 mm/day on average) in the RE model version. Interception losses were negligible in both model versions (< 4 mm/y). Woody vegetation cover was strongly reduced in the updated model version, resulting in a higher grass cover and a lower total vegetation cover overall (**Figure S3**). Especially evergreen trees, being the woody PFT with the highest cover (7.6%) in the default model version saw a strong reduction to 0.6% in the RE-based version. The overall lower vegetation cover resulted in a higher fraction of bare soil (42.0% in the new model), leading to an overall higher soil evaporation rate (**Figure 2**b). Our new model version also simulated an overall reduced surface runoff during rainfall peaks, from averages of 61.9 mm/y for the default model to 57.1 mm/y for the RE-based model (**Figure 2**c). Also baseflow runoff was reduced (2.31 mm/y for default model to 1.85 mm/y for RE-based model), while lateral flow had an increase over the entire rainy season, compared to the default model version (from 3.05 mm/y for the default model to 11.1 mm/y for the RE-based model).

Including the other boundary conditions, our model versions had a very distinct average seasonal cycle in soil moisture content (**Figure 3**). The default model hydrology simulated a build-up of soil moisture with depth for both the shallow (0–50 cm) and the deep (50–150 cm) soil layers, except for the bottom soil layer which was drained. Simulated soil moisture showed a clear discontinuity around 50 cm soil depth, both in magnitude and timing (**Figure 3a**). In contrast, the RE-based model versions





simulated a continuous soil moisture profile, decaying with layer depth and following the timing of the rainy season (**Figure 3b-d**).

We found little difference between the "free drainage" and "bedrock" bottom boundary conditions (**Figure 3b-c**). However, the "aquifer" boundary condition showed a strong upward capillary movement of water from the imposed saturated layer below the bottom layer. This resulted in a year-round soil moisture availability below 60 cm depth, with values higher than the soil water added by the rainy season precipitation (**Figure 3d**). The impacts of this aquifer on the simulated vegetation cover are discussed in detail further in this paper (Section 3.5) where we show the results of GWTD sensitivity tests.

Evaluating both model versions (with free drainage) against in-situ measurements of carbon and water fluxes showed only small differences in model performance, for both the full-year and wet-season metrics (**Figure 4**, **Table 1**). Simulated GPP by both model versions underestimated the eddy covariance-derived GPP overall. The start of the growing season around day 180 was much more abrupt in both models, compared to the smoother transition in the measurements. This is because all raingreen plants become active with full leaf cover at the same time in the model, while in reality there will be some variability in leaf flushing between individuals, as well as a more gradual leaf growth. Dry season GPP fluxes were underestimated by both models and close to zero overall, although the default model version had a larger tail in GPP after the end of the rainy season (**Figure 4**a). Ecosystem respiration fluxes (Reco) during the rainy season were slightly better represented in the updated model version, having lower RMSE and higher correlation with observations than the default version, while dry season respiration was relatively well captured by both models (**Figure 4**b, **Table 1**). As the sum of these components, the total NEE was poorly represented by both models, underestimating the observed net productivity in the rainy season and simulating a small source of $CO_2$ during the dry season, in contrast to the observed sink (**Figure 4**c). In contrast to carbon fluxes, water fluxes (E) were fairly well represented by both models (**Figure 4**d, **Table 1**). The new RE-based model showed a better agreement with the peak evaporation during the wet season, while slightly overestimating the early wet season E and underestimating its tail. Similar to its GPP performance, the default model version better captured the fluxes at the start of the dry season (**Figure 4**d). Water use efficiency was underestimated by both models during the core growing season, mostly due to the underestimation of GPP. However, simulated WUE values were close to the measuements during the first month of the growing season. On the other hand, the low ET values in the late growing season and the dry season let to a significant overestimation of WUE in those periods (**Figure 4**e). For both models, simulated WUE was the closest to observation-derived WUE during the wet season, where the default model had a lower RMSE, but the RE-based model had a sligthly higher correlation (**Table 1**). For reference, full time series are shown in the supplementary materials (**Figure S4**).

Soil moisture in the two upper layers (5cm, 10cm) was overestimated by the RE-based model, while the default version underestimated soil moisture in these layers (**Figure 5**a,b). The best agreement with observations was found at 30cm soil depth, where both models captured the soil moisture relatively well during the rainy season (**Figure 5**c). For the deeper layers the models start to diverge from each other and from observations. At 50 cm depth the soil moisture peak by the start of the rainy season is relatively well captured by the default model, although soil moisture content is underestimated during the rainy season and overestimated in the early dry season. In contrast, the RE-based version shows a delayed rise in soil moisture, but



captures the tail of the rainy season and the start of the dry season relatively well (**Figure 5**d). At 100 cm depth neither model

version reproduces the observed soil moisture. The RE-based version underestimates – and negatively correlates with – measurements, while the default model version overestimates soil moisture and its peak is delayed by about 20 days. (**Figure 5**e, **Table 1**). While observed soil moisture still clearly shows a peak caused by the rainy season at 100 cm depth, both model versions have lost most of the rainy season fingerprint at this depth, especially its timing. Full time series are provided in the supplementary materials (**Figure S5**).

Finally, all model versions showed a good water mass balance closure, as the sum of the rainfall, evaporation and runoff components each year nearly matched the change in water column storage, resulting in an overall low accumulated water balance error (Eq. (13), **Figure S6**). The aquifer bottom boundary condition showed a higher water mass balance error than the other model setups, with an accumulated error equal to 0.14% of the accumulated rainfall over the years 2002–2022, while the other model versions performed better (< 0.002%) (**Figure S6**).




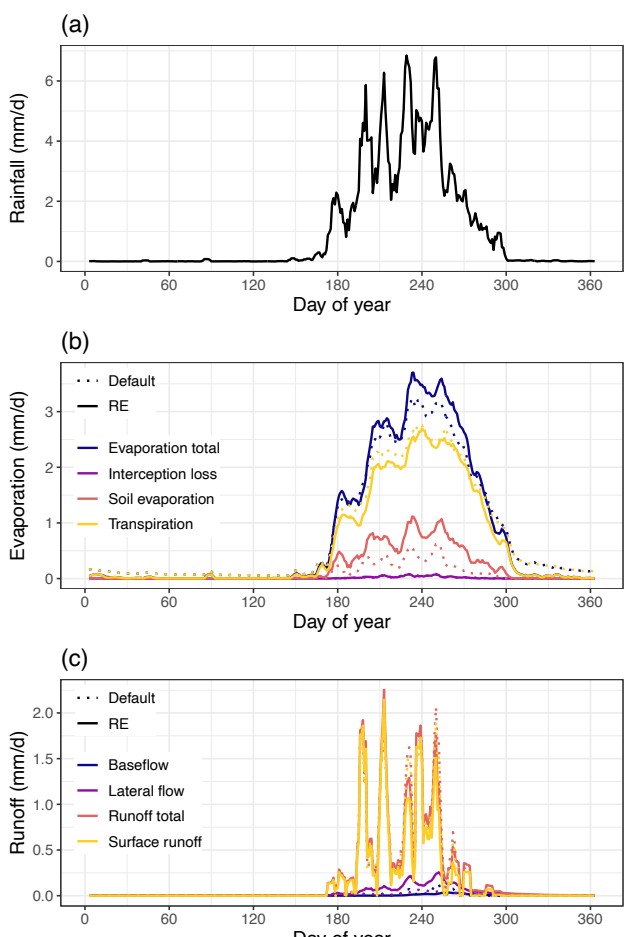

**Figure 2.** Average yearly cycle of simulated hydrology at the Dahra site in Senegal for 2002-2022, comparing the default LPJ-GUESS v4.1 multilayer soil hydrology (Default; dotted line) with the updated soil hydrology based on Richard's equations with the "free drainage" bottom boundary condition (RE; solid line). Figures show 5-day moving averages of (a) in-situ measured daily rainfall for reference, (b) simulated daily evaporation components, and (c) simulated daily runoff components.



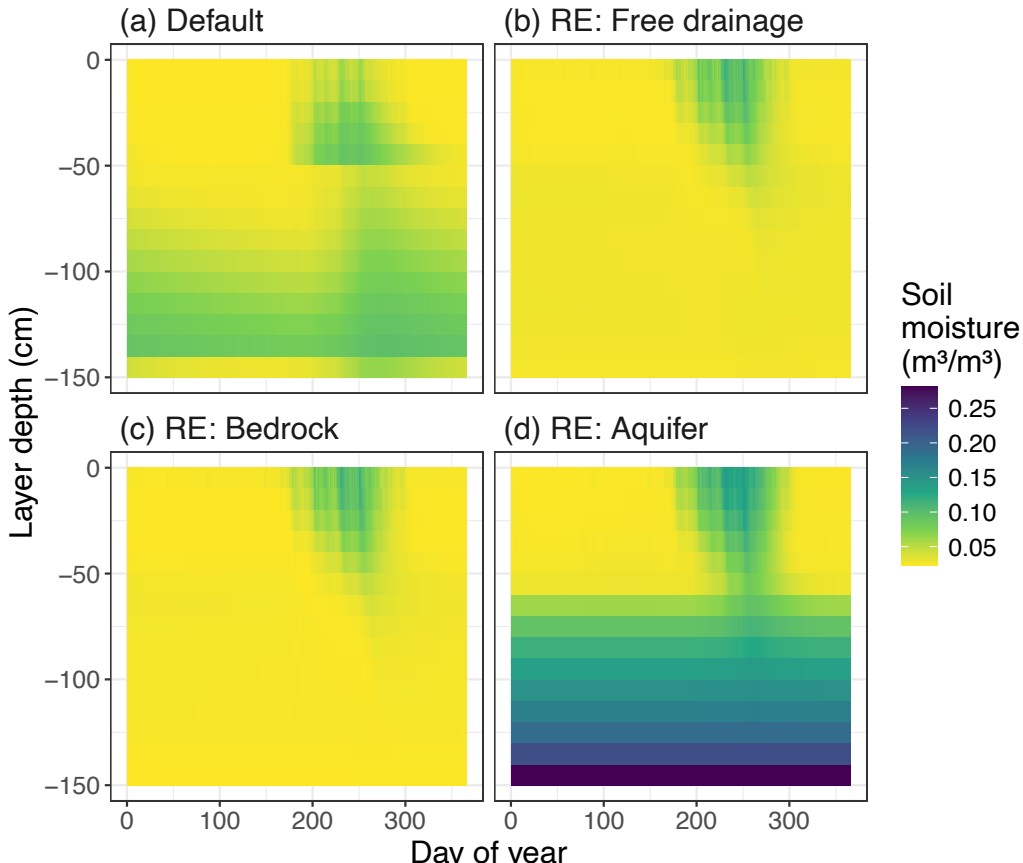

**Figure 3.** Average yearly cycle of soil moisture (m³/m³) for the Dahra fluxtower site, as simulated by the different soil
hydrology modules in (a) the default version of LPJ-GUESS v4.1 and (b-d) the updated version based on Richards equation
with the available bottom boundary conditions. Boundary conditions include (b) free drainage (the default used for model
evaluations), (c) impermeable bedrock, and (c) a permanent aquifer below the bottom layer.



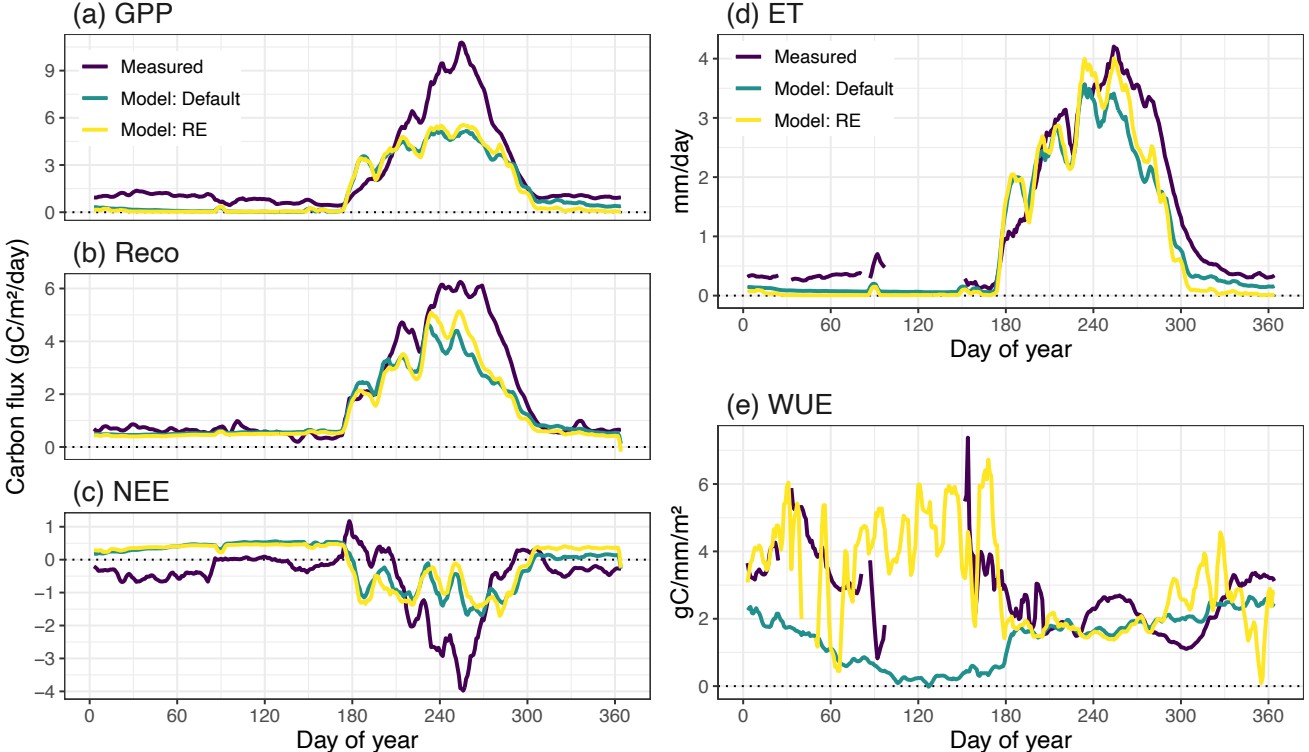

**Figure 4.** Average yearly cycle of measured vs. simulated *(a–c)* carbon, *(d)* water fluxes, and *(e)* water use efficiency at the Dahra site in Senegal for the period 2010–2020. Both the standard ("Default") and the updated ("RE") model versions of LPJ-GUESS v4.1 are compared against measurements of *(a)* GPP, *(b)* Reco, *(c)* NEE and *(d)* ET, as well as the derived WUE *(e)*. Figures show 5-day moving averages.



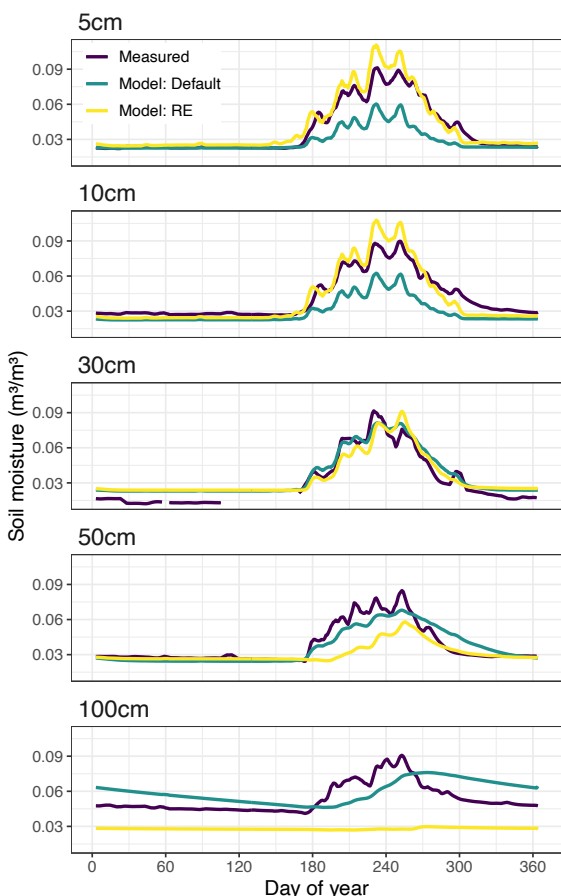

**Figure 5.** Average yearly cycle of measured vs. simulated volumetric soil moisture content at the Dahra site in Senegal for 2002–2022. Panels show 5-day moving averages of results at different soil layer depths (5–100cm). Simulated results were interpolated to match observed layer depths.


**Table 1.** Numerical evaluation of the the standard ("Default") and updated ("RE") versions of the LPJ-GUESS v4.1 model against measurements made at the Dahra site in Senegal. Carbon and water fluxes span the period 2010–2020 while soil moisture measurements (5–100cm) cover 2002–2022. Metrics used are the RMSE and Pearson correlation coefficient (R), calculated over the entire time-series for both the complete years as well as the rainy seasons separately. The start and end of 440 the rainy season is based on the climatological anomalous accumulation. All correlations are significant ($p < 10^{-3}$) except where indicated in bold with a cross (×). Units for RMSE values are $gC/m^2/day$ for carbon fluxes (GPP, Reco, NEE), mm/day for ET, and $m^3/m^3$ for soil moisture.

| Variable | RMSE | | R | |
|---|---|---|---|---|
| | Full year | Rainy season | Full year | Rainy season |





|  | Default | RE | Default | RE | Default | RE | Default | RE |
|---|---|---|---|---|---|---|---|---|
| GPP | 2.71 | 2.70 | 3.94 | 3.90 | 0.799 | 0.785 | 0.528 | 0.519 |
| Reco | 1.87 | 1.73 | 2.46 | 2.32 | 0.754 | 0.785 | 0.369 | 0.487 |
| NEE | 1.75 | 2.06 | 2.50 | 2.98 | 0.397 | 0.295 | 0.318 | 0.202 |
| ET | 0.898 | 0.983 | 1.00 | 1.20 | 0.869 | 0.869 | 0.718 | 0.725 |
| WUE | 2.24 | 2.28 | 0.98 | 1.31 | -0.204 | **-0.00$^{×}$** | 0.124 | 0.133 |
| SM 5cm | 0.0224 | 0.0235 | 0.0367 | 0.0397 | 0.767 | 0.804 | 0.711 | 0.699 |
| SM 10cm | 0.0211 | 0.0217 | 0.0326 | 0.0357 | 0.788 | 0.814 | 0.725 | 0.707 |
| SM 30cm | 0.0176 | 0.0236 | 0.0229 | 0.0349 | 0.799 | 0.668 | 0.568 | 0.356 |
| SM 50cm | 0.0166 | 0.0258 | 0.0200 | 0.0375 | 0.711 | 0.435 | 0.547 | 0.275 |
| SM 100cm | 0.0277 | 0.0337 | 0.0312 | 0.0509 | 0.097 | -0.138 | 0.194 | -0.130 |

## 3.2 Regional performance

In line with the MAP levels, the simulated total evaporation (sum of plant transpiration, bare-soil evaporation and interception loss) of the RE-based LPJ-GUESS model showed a strong north-south gradient across the entire region, when averaged over the period 1980–2022 (**Figure 6a**). Evaporation values ranged from 100 mm/y in the north to 960 mm/y in the southern parts. The updated model matched the GLEAM total evaporation dataproduct relatively well: the average difference between the model and GLEAM over all gridcells was -6.2 mm/year, with a spatial standard deviation of 93.7 mm/year (**Figure 6b**, **Figure**
**7a**). The strongest underestimations by the model were found near water bodies (e.g. along the Senegalese coastline) as the GLEAM total evaporation also includes open water evaporation. The evaluation also showed strong spatial gradients, as the model output tended to lower than GLEAM evaporation in the northern and western parts, while being higher in the southern parts and east of 15ºE (**Figure 6b**). The correlation between our model and GLEAM over 1980–2022 was overall positive, with a few negative correlations in the eastern and western parts of the region. The average correlation was 0.34 with a standard
deviation of 0.26 (**Figure 6c**, **Figure 7b**). The performance of the evaporation simulated by the default model was very close to that of the updated model version, showing nearly identical model-GLEAM differences and correlations (**Figure 7a,b**). Similar to the total evaporation, averages of yearly simulated plant transpiration also followed a strong north-south gradient across the region, with values ranging from 32 mm/y to 880 mm/year (**Figure 8a**). Our RE-based model tended to simulate higher transpiration rates than GLEAM, especially in the eastern parts of the region, with a spatially averaged value of 48
mm/y and a spatial standard deviation of 71 mm/y (**Figure 8b, Figure 7c**). The new model version performed better than the default version, as the latter simulated transpiration rates that were 108±61 mm/y higher than GLEAM (**Figure 7c**). The correlations between either model version and the GLEAM transpiration time-series over 1980–2022 were overall positive,



but low on average (R=0.25 for both models) due to a large spatial variability in the correlation, similar to the spatial pattern of correlations of the total evaporation (**Figure 8c, Figure 7d**).

Complementary to the general overestimations of simulated transpiration, time-averaged bare-soil evaporation was overall underestimated (-59±61 mm/y) by the RE model across most of the region, except in the southeastern part (**Figure 8d,e**). Correlations between the model and GLEAM time series were overall positive (R=0.19±0.24) and followed a strong north-south gradient, where the gridcells showing the highest correlations were found in the northern parts, and most gridcells with a negative correlation were found in the south (**Figure 8f**). Soil evaporation simulated by the RE-based model performed better

than the default model, showing overall smaller differences (Default: -96±51 mm/y) and larger correlations (Default: 0.13±0.2) with the GLEAM soil evaporation dataproduct (**Figure 7e,f**).

Simulated surface and root-zone soil moisture showed very similar geographical patterns, with overall lower soil moisture values in the northern areas of the region, especially around 10ºE, while the highest soil moisture values were found in the southern and eastern areas (**Figure 9a,d**). Yet, the RE-based model tended to overestimate GLEAM soil moisture in the

northern parts, while underestimating it in the southern half, with an average error of -0.012±0.047 $m^3/m^3$ for surface soil moisture and -0.025±0.050 $m^3/m^3$ for root-zone soil moisture (**Figure 9b,e**). Correlation with GLEAM timeseries were mostly positive throughout the entire region (R=0.52±0.21 for surface SM, R=0.42±0.20 for root-zone SM) (**Figure 9c,f**). Simulated surface soil moisture by the RE-based model performed better than the default model version, showing both a smaller error and a better correlation with the GLEAM dataproduct (**Figure 7g,h**), while the performance of simulated root-zone soil

moisture was comparable between both models (**Figure 7i,j**). Similar to the result for the Dahra site, the average yearly cycle of the soil moisture profile (here additionally averaged over all gridcells) was highly distinct between the old and the new model versions, showing similar results as for the Dahra site-level study (**Fig S4**).



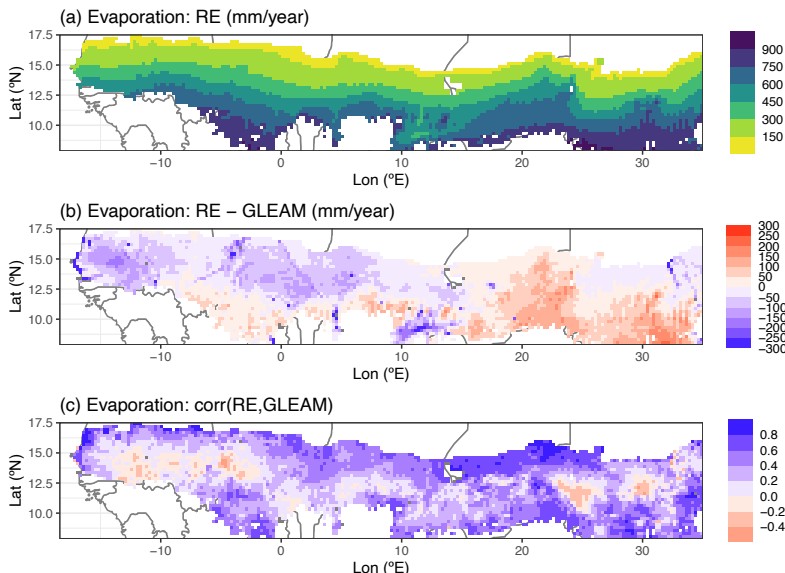

**Figure 6.** Regional simulation of yearly total evaporation by the updated LPJ-GUESS v4.1 model for the period 1980–2022. (a) Multiyear model averages over this time period, (b) difference between multiyear averaged model and GLEAM data, (c) Pearson correlation between model and GLEAM for yearly totals over 1980–2022. For panel (b) all outlier values saturate at ±300 mm/year to avoid distorting the colour scale.



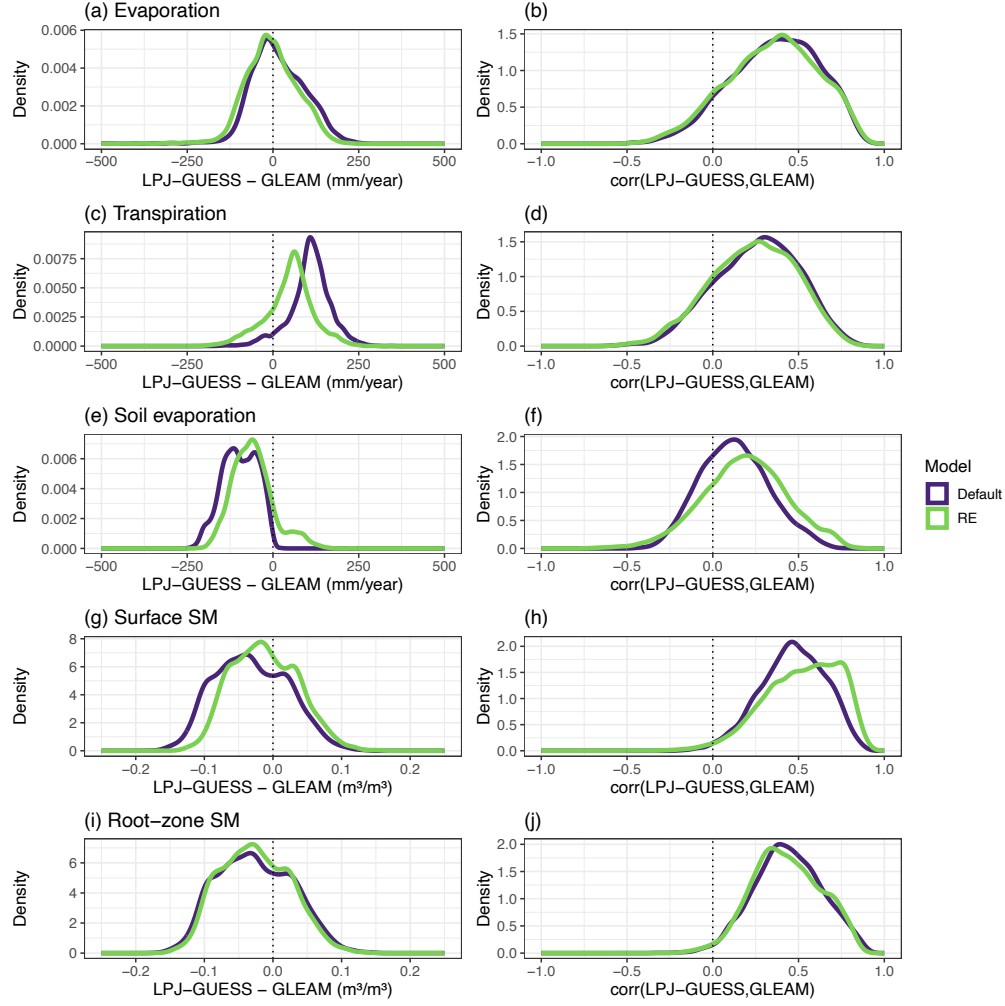

**Figure 7.** Comparison of the hydrology performance of the standard (Default) and updated (RE) versions of the LPJ-GUESS model against the GLEAM dataproduct. The left column shows the dustribution of time-averaged differences between the model and GLEAM over all gridcells, while the right column shows the distruibution of corelations with GLEAM, both for the period 1980-2022. The evaluated variables are yearly (a,b) total evaporation, (c,d) transpiration, (e,f) soil evaporation, (g,h) surface soil moisture, and (i,j) root-zone soil moisture.



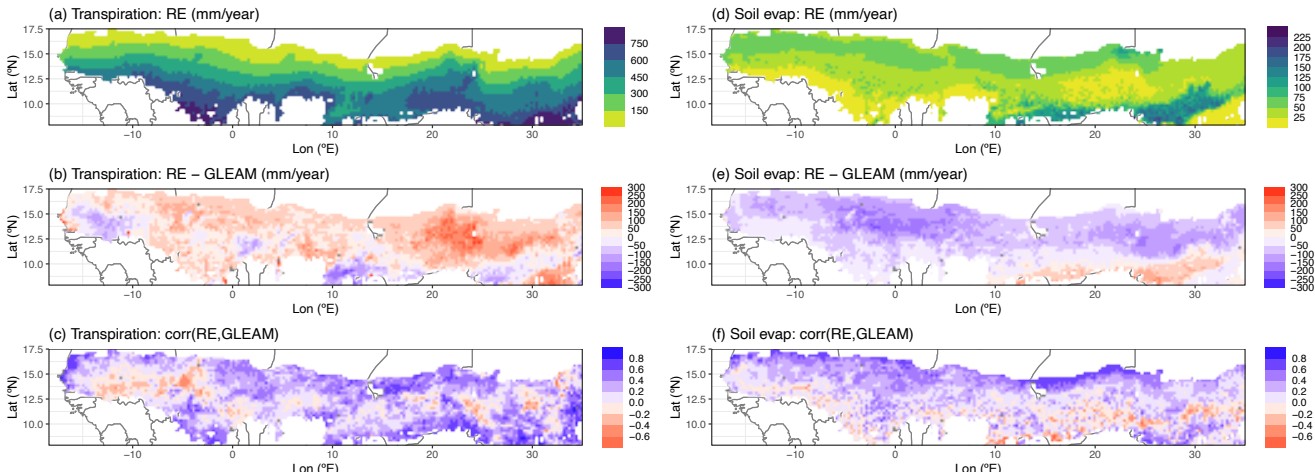

**Figure 8.** Regional simulation of yearly total transpiration (left) and soil evaporation (right column) by the updated LPJ-
GUESS v4.1 model for the period 1980–2022. (a,d) Multiyear averages over this time period, (b,e) difference between
multiyear averaged model and GLEAM data, (c,f) Pearson correlation between model and GLEAM over 1980–2022. For
panels (b) and (e) all outlier values saturate at ±300 mm/year to avoid distorting the colour scale.

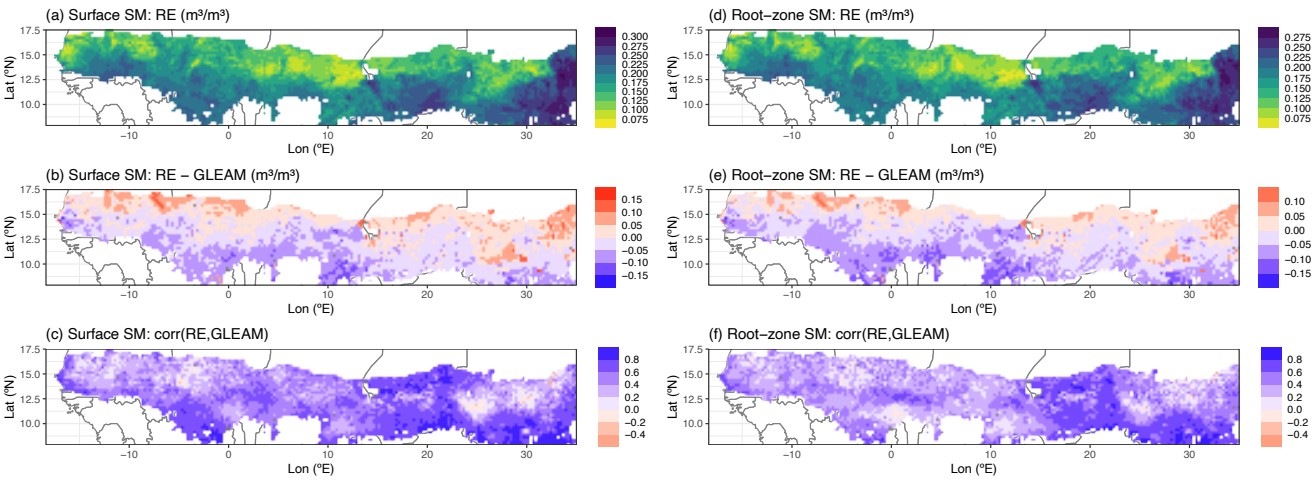

**Figure 9.** Regional simulation of surface soil moisture (left) and root-zone soil moisture (right column) by the updated LPJ-
GUESS v4.1 model for the period 1980–2022. (a,d) Multiyear model averages over this time period, (b,e) difference between
multiyear averaged model and GLEAM data, (c,f) Pearson correlation between model and GLEAM over 1980–2022. For
panels (b) and (e) all outlier values saturate at ±0.25 m³/m³ to avoid distorting the colour scale.




## 3.3 Impacts on simulated regional vegetation cover

Changing the soil hydrology in LPJ-GUESS had a large impact on simulated vegetation cover across the entire Sudan-Sahel region (**Figure 10**, **Figures S8-S12**). Average total vegetation leaf area index (LAI) over the period 2000–2022 decreased over the arid parts of the Sahel but increased in the more humid parts after implementing the model updates, amounting to an overall average increase of 0.204 m$^2$/m$^2$ (**Figure 10**). Both model versions overestimated the averaged MODIS LAI measurements, but remained within the 5-95 percentile range over all gridcells in each MAP bin (**Figure 10**). These spatial patterns were

mainly driven by changes in C$_4$ grass LAI (+0.47 m$^2$/m$^2$ overall) and tropical shrub LAI (+0.39 m$^2$/m$^2$) which followed the overall north-south gradient in LAI changes (**Figures S9–S10**). Especially shrub cover increased significantly in the southern parts, with changes in LAI up to 3.32 m$^2$/m$^2$ for southern Chad and Sudan (**Figure S10**). This shrub cover mostly replaced raingreen tree PFT which decreased overall (-0.48 m$^2$/m$^2$) and especially in southern Chad and Sudan with reductions in LAI down to -2.75 m$^2$/m$^2$ (**Figures Figure 10**, **S10**). Evergreen tree LAI also decreased overall (-0.15 m$^2$/m$^2$) in the new model

version but saw a slight increase in southern Chad and Sudan (**Figures Figure 10, S11**).

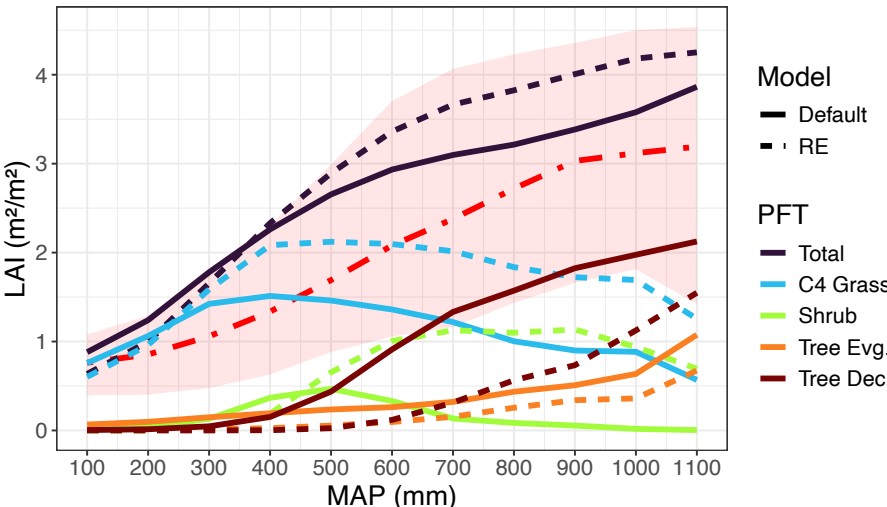

**Figure 10.** Simulated vegetation LAI (yearly maxima) in function of MAP over the Sudan-Sahel region for both versions of the LPJ-GUESS model. Model output and rainfall data were averaged over 2000–2022 and further averaged into MAP bins of

100 mm. The red dash-dotted line and the red shaded area represent the MODIS LAI data (2000-2022) average and 5-95 percentile range over all MAP bins, respectively.

## 3.4 Soil texture sensitivity

In this section we present the results of our soil texture sensitivity analysis for which we used the meteoroiological drivers from the Dahra fluxtower site, but artificially varied the soil texture across all possible sand-silt-clay combinations.





Focusing on the RE-based model, we found that soil texture had a significant influence on the multiyear averaged simulations of the evaporation components (**Figure 11**, **Table 2**). Total evaporation mainly varied with clay content for the new model, ranging from ~320 mm/year for low-clay soils to 183 mm/y on clay soils (**Figure 11**, **Figure S13**). Interception loss was similarly dependent on clay content, with a preference for more silty soils when clay content was low, while soil evaporation was mostly dependent on silt content (48–73 mm/y). Transpiration showed a similar pattern as total evaporation, being its

main contributor. Transpiration was highest (268 mm/y) for low-clay soils for higher silt content (> 60% silt) and lowest (112 mm/y) for clayey soils (**Figure 11**, **Figure S13**). Total evaporation, transpiration and interception loss were highly correlated (R>0.94, p<0.0001) with plant-available water capacity ($\theta_{awc}$), while soil evaporation showed an anti-correlation with $\theta_{awc}$ (R=-0.65, p<0.0001) (**Figure S14**). The correlation between the evaporation components and the shape parameter $b$ was also high and of the opposite sign, whereas total evaporation, transpiration and interception loss had a strong anti-correlation (R<-

95, p<0.0001) and soil evaporation had a high correlation (R=0.66, p<0.0001) (**Figure S14**). Other hydraulic parameters, such as $\theta_{wp}$ and $\theta_{fc}$ were either related to $\theta_{awc}$ or had a low (Ksat) or insignificant ($\theta_{sat}$) correlation with evaporation (**Figure S14**). Surface runoff was the highest contributor to total runoff, and was strongly anti-correlated (R=-0.96, p<0.0001) with $\theta_{awc}$, as soils with a high soil water capacity can absorb more water before they saturate, leading to lower surface runoff rates (**Figure S14**).

The marked influence of soil texture on soil hydrology was coupled with a similar influence on simulated vegetation cover and composition (**Figure 12**). In the RE-based model, $C_4$ grass LAI mainly varied with soil clay content, leading to a higher $C_4$ grass cover on soils that were low in clay content, while woody vegetation LAI was highest for silt-rich soil types (silt, silt loam, silty clay loam) and was significantly lower (near zero) over all other soil types (**Figure 12**, **Figure S15**). Total vegetation LAI had a strong correlation (R=0.95, p<0.0001) with plant-available water capacity $\theta_{awc}$ and a strong anti-correlation (R=-

0.98, p<0.0001) with the shape parameter $b$. Both were mainly driven by $C_4$ grass LAI, as woody vegetation cover was overall low for the RE-based model (**Figure S16**).

Comparing both models revealed an overall higher sensitivity to soil texture in the RE-based model than the default model (**Table 2**). The coefficient of variation of evaporation and vegetation cover over all soil textures was higher in the RE-based model than the default model for all variables, except for soil evaporation (**Table 2**). This was also reflected in the ternary

plots of both variables, which showed an overall higher variation for the RE-based model, as well as more defined (less patchy) patterns of variation with soil texture (**Figure 11**, **Figure 12**). Simulated hydrology components by the new model also had a higher range and steeper slope to variations in $\theta_{awc}$ and $b$, compared to the default model (**Figure S14**). For simulated vegetation cover this was only the case for total LAI and $C_4$ grass LAI, as woody vegetation LAI was overall lower in the RE-based model (**Figure S15**). Overall, the simulated vegetation differences between the RE-based model and the default model resembled

those from the reference Dahra simulations shown earlier.

As expected, sandy soil types resulted in the lowest error (RMSE) between simulations and measurements of soil moisture from the Dahra site, as evaluated for each soil layer independently (**Figure S17**).





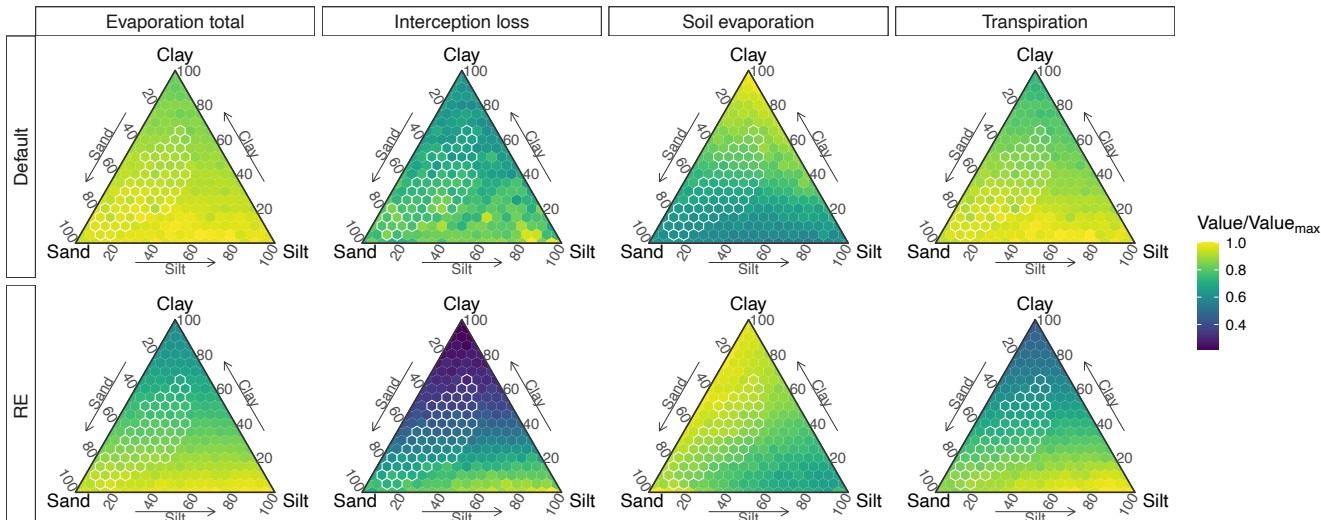

**Figure 11.** Simulated sensitivity of evaporation components (columns) to soil texture for both model versions (rows), based on the Dahra meteorological drivers. Sensitivities represented by ternary plots of time-series average evaporation component values for each soil type. Values are divided by maximum evaporation value over all soil types (Value$_{max}$), calculated for each model version and each evaporation component separately. Soil textures found in the Sudan-Sahel region are marked in white.

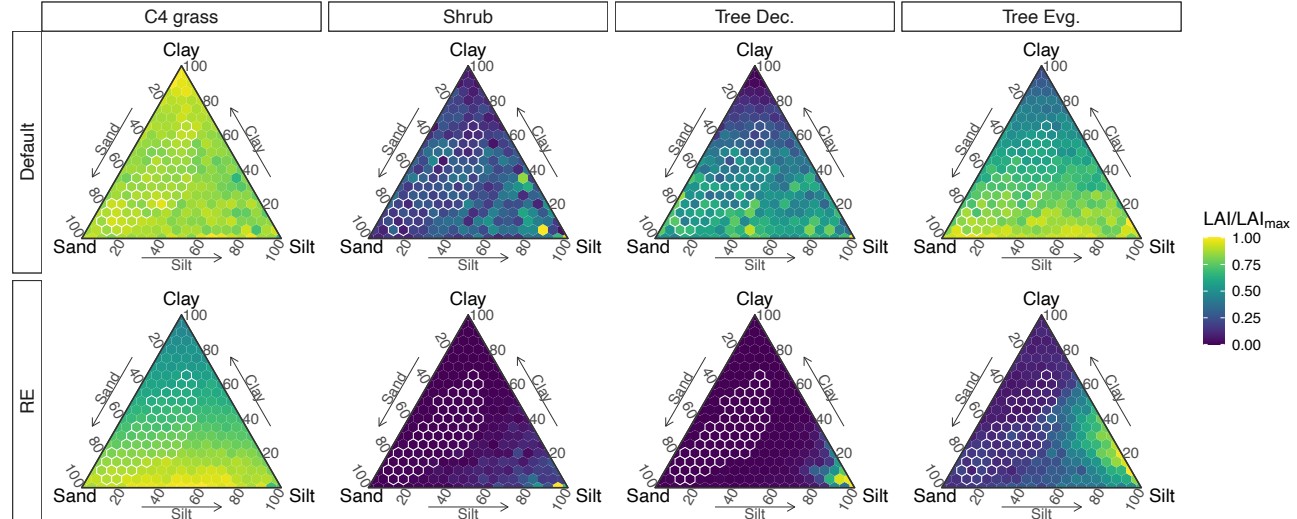

**Figure 12.** Simulated sensitivity of the different dryland PFTs (columns) to soil texture for both model versions (rows), based on the Dahra meteorological drivers. Sensitivities represented by ternary plots of time-series average LAI for each soil type. Values are divided by the maximum LAI value over all soil types (LAI$_{max}$), calculated for each model version and each PFT separately. Soil textures found in the Sudan-Sahel region are marked in white.





**Table 2.** Mean, standard deviation (SD) and coefficient of variation (CV) of different time-averaged evaporation components and vegetation LAI. Statistics calculated over all soil texture combinations, for a site-level simulation based on the Dahra meteorological drivers. Units refer to mean and SD, while the coefficient of variation (CV=SD/mean) is unitless.

| Model output | | Mean | | Standard deviation | | Coefficient of variation | |
|---|---|---|---|---|---|---|---|
| | | Default | RE | Default | RE | Default | RE |
| Evap. (mm/y) | Evaporation total | 295 | 267 | 13.9 | 34.9 | 0.047 | 0.131 |
| | Interception loss | 3.47 | 2.42 | 0.383 | 0.872 | 0.110 | 0.360 |
| | Soil evaporation | 35.8 | 62.0 | 5.84 | 6.73 | 0.163 | 0.109 |
| | Transpiration | 256 | 202 | 19.3 | 39.2 | 0.075 | 0.194 |
| LAI ($m^2/m^2$) | Total | 1.80 | 1.630 | 0.088 | 0.324 | 0.049 | 0.198 |
| | $C_4$ Grass | 1.47 | 1.570 | 0.088 | 0.301 | 0.060 | 0.192 |
| | Shrubs | 0.08 | 0.021 | 0.043 | 0.047 | 0.538 | 2.20 |
| | Tree Deciduous | 0.07 | 0.003 | 0.032 | 0.012 | 0.415 | 3.39 |
| | Tree Evergreen | 0.15 | 0.012 | 0.038 | 0.010 | 0.255 | 0.812 |

### 3.5 Groundwater table depth influence

Here we show the modelled sensitivity of dryland vegetation to GWTD, for which we used the RE-based model with the "aquifer" bottom boundary condition at various depths. Changing the bottom boundary condition to represent an ground water layer below the bottom layer already had a large impact on the soil water profile for the default soil depth (150 cm) (**Figure 3**). Varying groundwater location over a range of depths (75–600 cm) had a large impact on simulated vegetation cover and surface hydrology, as total vegetation cover was sensitive to GWTD variations down to 200 cm, while vegetation composition became nearly insensitive to GWTD below 350 cm (**Figure 13**). Simulated tropical evergreen tree cover was largely impacted by this change in water availability, as these deeper-rooted tree PFTs now had access to an unlimited water supply. This PFT could photosynthesize and transpire throughout the year, and became the main vegetation cover with an LAI of more than 3 $m^2/m^2$ (**Figure 13a**). This effect became even more pronounced for shallower groundwater depths, increasing evergreen tree LAI up to 4.5 $m^2/m^2$ for a groundwater depth of 75 cm. On the other hand, for simulations with a deeper groundwater table, the evergreen tree LAI declined rapidly with groundwater depth, and $C_4$ grass became the dominant vegetation cover again for water table depths below 225 cm (**Figure 13a**).

Evaporation components followed this relationship with groundwater depth, and total evaporation was again mainly driven by transpiration during both seasons (**Figure 13b**). During the rainy season, total evaporation was invariant (~2.4 mm/day) to





GWTD for levels below 200 cm. Plant transpiration accounted for 75% of total evaporation, while the remainder came from bare soil evaporation. Rainy season total evaporation increased sharply for shallower GWTD, with an increasing contribution by transpiration (up to 90%) due to the higher vegetation cover. During the dry season, all evaporation was due to transpiration, as no surface soil water was available for evaporation. Transpiration rates were near zero for GWTD below 250 cm, but again increased sharply for more shallow groundwater depths, up to rates that matched rainy season transpiration (**Figure 13b**).

Surface runoff and lateral flow runoff components were not influenced by groundwater depth, except for baseflow runoff, which was negative as water entered the soil through the bottom layer, rather than exiting the soil (**Figure 13c**). This baseflow "runon" component had the same relationship to groundwater depth as transpiration, for both seasons. For groundwater depths below 200 cm the baseflow rate varied only little with groundwater depth. However, for more shallow groundwater depths the baseflow runon rates increased again sharply to compensate for the soil water lost by transpiration (**Figure 13c**).

Transpiration during the rainy season mainly occured from the upper soil layers, peaking near the surface and declining down to ~50 cm for all simulated groundwater depths. However, during the dry season, transpiration shifted to deeper soil layers, peaking between 85–150 cm for all GWTDs deeper than 200 cm, and at more shallow layers for GWTD above 200 cm (**Figure 14**). For groundwater depths above 300 cm there was also a peak in transpiration from the bottom soil layer. This peak grew in importance for more shallow groundwater depths, reaching 25% of all transpiration for simulations where groundwater was located more shallow than 150 cm. Note that the values in **Figure 14** are not weighted by layer size, and larger layers will naturally contribute a higher percentage due to their size (see **Figure S18** for a weighted version).

All of these results can be explained by intersecting the soil water profile with the root distribution over all layers. Soil water content increased with layer depth during the dry season (**Figure S19**), while root distribution decreased (**Figure S20**), leading to a local maximum in root water uptake where both functions cross. However, for soil depths of 200 cm and less, an increasing amount of root biomass is assigned to the bottom layer in LPJ-GUESS (see Methods). Especially for soil depths more shallow than 100 cm, the bottom layer will host the the largest fraction of tree PFT root biomass. Therefore, for shallow soil depths (which the model originally was not designed for) this buildup of root biomass in the bottom layer further amplfied the contribution to transpiration of this layer, due to its high soil water availability.

Finally, re-evaluating the yearly cycle of soil moisture against site-level measurements showed that introducing a groundwater table did not significantly improve the match with observed soil moisture content for the sampled depths at the Dahra site, for any of the simulated GWTDs (**Figure S21**).





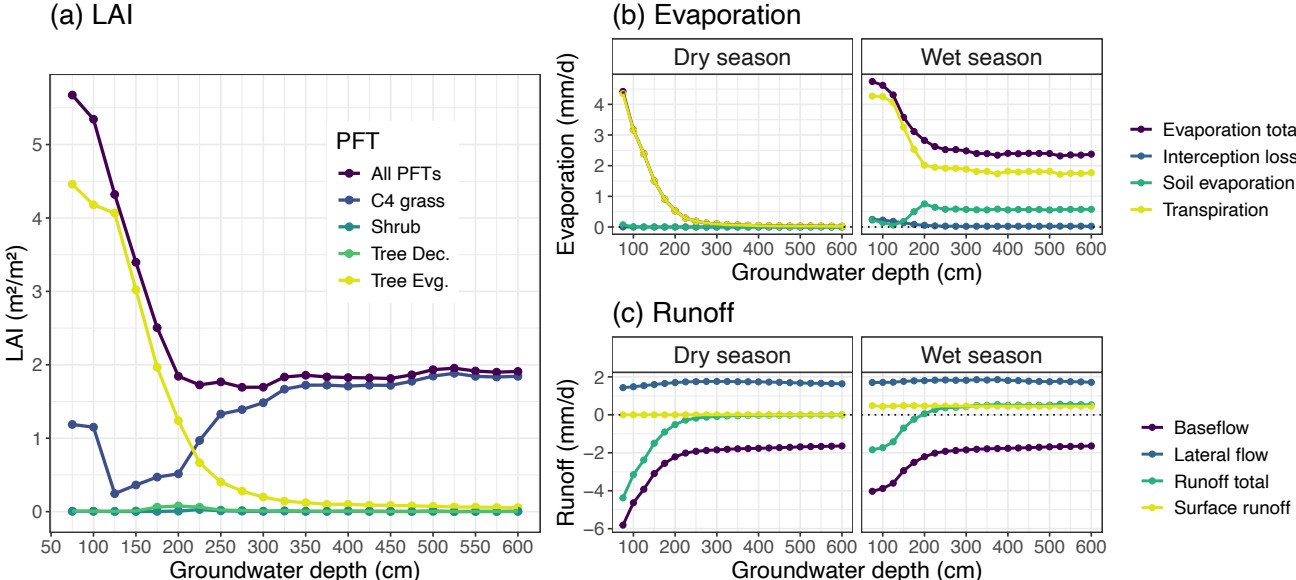

**Figure 13.** Influence of groundwater depth on simulated vegetation cover and surface hydrology at the Dahra flux tower site. Results are averaged over all simulated years (2002–2022) and further separated into the dry and wet season for the surface hydrology results. Panels include (a) vegetation cover as given by the LAI of the different PFTs, (b) evaporation components and (c) runoff. All results in function of separate simulations of various groundwater depths.




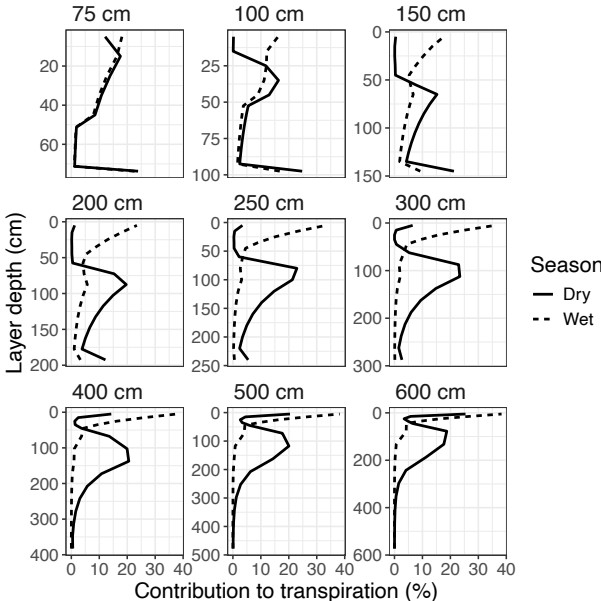

**Figure 14.** Contribution of each soil layer to the simulated total plant transpiration at the Dahra flux tower site, for a selection of ground water depths, ranging from 75–600 cm (panels). Results are averaged over all simulated years (2002–2022) and further separated into the dry and wet season. Results are not weighted by soil layer size.

## 4 Discussion

Based on ecosystem-level evaluations against site-level measurements and regional data products, updating the soil water dynamics in LPJ-GUESS resulted in an overall lower bias and higher correlation with observations, although the improvements were only small overall. This indicates that other factors limit overall ecosystem-level model performance for this region. Given the high human population in the Greater Sahel, anthropogenic factors such as land use (change) and animal grazing can have a large impact on the ecosystem, yet are not accounted for in the model (Brandt et al., 2017; Lindeskog et al., 2013; Souverijns et al., 2020). In our simulations we also did not account for fire, which would further increase the realism of the model (Axelsson and Hanan, 2018; Sankaran et al., 2008). For the model evaluation, we only used the "free drainage" boundary condition with a fixed soil depth for the updated model. As we showed, including a groundwater table and varying soil depth can have a large impact on the simulated vegetation and associated fluxes, especially for soil depths shallower than 300 cm. Including these boundary conditions, e.g. based on maps of water table depth, may therefore further improve the overall model performance without further changes to the soil representation itself.

The default LPJ-GUESS model already underestimated dry season transpiration and photosynthesis, and this is even more so for the updated model version, for which these fluxes are nearly zero. This can be explained by the lower simulated tree cover. Both tree PFTs were overall strongly reduced in the new model version, causing sharper decline in total transpiration after the



rainy season ends, which is now mostly driven by more shallow-rooted grass. The timing of plant transpiration therefore mainly follows the surface soil moisture availability, driven by the timing of the rainy season. Nevertheless, we also showed that dry

season transpiration can be largely influenced by groundwater access in the new model version, simulating values that well exceed fluxtower measurements when assuming an (unrealistic) shallow water table depth.

Our results also clarify why the default model version supports evergreen trees in drylands, as the year-round high availability of soil water at deeper soil layers (down to -1.4 m) in this model enables these PFTs to stay productive during the dry season. However, it is highly questionable whether this simulated soil water availability is realistic, as we showed that a more advanced

soil percolation scheme does not support such a layer. Nevertheless, even the original percolation function of the default model would not suggest to support such a layer either. The answer lies in the model code, which contains a condition that percolation can only occur during days for which the sum of rainfall and snow melt is higher than 0.1 mm (Nord et al., 2021). This condition causes any buildup of deep soil moisture during the wet season to remain in place until the next rainy season, while slowly being depleted by evergreen tree transpiration. Our new model version does not contain such a condition and therefore

does not suffer from this issue. Therefore, the default version of LPJ-GUESS should be used with caution when simulating vegetation in regions with a highly seasonal climate, especially drylands.

Despite all this, our RE-based model improved the match with observed soil moisture only for the upper (30 cm) soil layers. For the deeper layers the footprint of the rainy season disappears too quickly in both model versions, as both models smooth out the changes in soil moisture much sooner than what is found in soil moisture observations. This suggest that the

pedotransfer functions (e.g. for calculating soil hydraulic conductivity) need to be fine-tuned for this region. These are still based on the original parameterization of Cosby et al. (1984) while more advanced versions have become available since then (Van Looy et al., 2017; Weber et al., 2024). We suggest that future model developments, of LPJ-GUESS as well as several other DVMs that still use the Cosby et al. (1984) parameterizations, take these into account (Meunier et al., 2022). Another option would be to use soil hydraulic traits from aggregated trait maps directly, rather than deriving them from aggregated soil

texture maps (Montzka et al., 2017). Nevertheless, other unaccounted processes can also play a large role in dryland soil moisture dynamics, such as hydraulic redistribution and water infiltration along roots (Barron-Gafford et al., 2017; Bogie et al., 2018; Wang et al., 2023). Our model also simplifies soil representation by assuming vertically homogeneous hydraulic properties in each soil column, which can also impact on soil water dynamics.

Our new developments open up the LPJ-GUESS model for simulating dryland ecohydrology more realistically. The

percolation scheme based on gradients in water potential improves the model's soil moisture dynamics, and it is largely based on physically measurable quantities (Bonan, 2019; Ireson et al., 2023). This scheme also allows water to move upwards against gravity, which becomes particularly important when simulating the aquifer bottom boundary condition. The option to simulate different soil depths and different bottom boundary conditions gives the model more flexibility.

This increased flexibility was highlighted by our sensitivity tests. Activating the aquifer bottom boundary condition and

varying the depth of this layer had a large impact on simulated vegetation cover and surface hydrology. These simulations suggest that dryland ecosystems like the one we studied here could shift from a groundwater-dependent (bottom supply) to a





rainfall dependent (top supply) ecosystem depending on changes in GWTD and root distribution (Fan et al., 2017; Rohde et al., 2024). Using the bedrock boundary condition could have a similar effect for ecosystems with higher rainfall and shallow soil depth, but we did not make any tests on the potential of this boundary condition. Our sensitivity test on soil texture also

revealed a higher overall sensitivity of total vegetation cover and $C_4$ grasses to soil texture, especially when the latter is translated into plant-available water capacity. This further opens up the model to simulate a higher range in dryland plant cover, which can also be enhanced by updating the pedotransfer functions (Meunier et al., 2022). Woody plants also had a larger year-to-year variability in vegetation cover in our new model version, but their sensitivity to soil texture is obscured by to the lower mean woody PFT leaf cover overall.

From a practical perspective, the new model code is relatively simple and easy to transfer to other branches of the LPJ-GUESS model, as well as to other vegetation models (Ireson et al., 2023). The model does not need any new parameters to function, and is flexible for specifying soil depth and using different bottom boundary conditions. Computing a solution to Richards equation is computationally relatively expensive. We did not perform any systematic benchmarks, but from our experience the model takes about 5 times longer to run with the new model hydrology. Using the aquifer bottom boundary condition further

increased this runtime by an additional factor 10, caused by the large gradient in soil moisture that requires shorter (i.e. more) subdaily timesteps. This also resulted in the larger water mass balance erorr for the aquifer boundary condition. Using a matrix-based approach to solving Richard's equations would be a way forward towards increasing the computational efficiency (Bonan, 2019).

The model development that we present here does not solve all challenges of LPJ-GUESS for simulating dryland ecohydrology

correctly. We identified at least two additional developments that need to be merged with with our model in order to simulate plant-water dynamics in arid ecosystems realistically. After improving the soil hydrology (LPJ-GUESS-DRY v1.0) the next milestone will implement a better representation of plant hydraulics. For dryland ecosystems it is important to have a highly resolved representation of drought stress and hydraulic dynamics through the soil-plant-atmosphere continuum (Medlyn et al., 2016; Xu et al., 2016). Fortunately, a new plant hydraulics scheme was recently developed for LPJ-GUESS, including drought-

induced mortality due to cavitation (Papastefanou et al., 2020, 2024). This model version has been shown to capture drought-induced vegetation mortality in the Amazon much more realistically than the default version (Papastefanou et al., 2024). Merging this development with our model version is foreseen in the next update of our model.

Another ecohydrological limitation for simulating dryland ecosystems with LPJ-GUESS stems from the static root architecture in the model, where root biomass is distributed based on an exponential equation (Jackson et al., 1996) and limited to a soil

depth of 150 cm. However, dryland vegetation is known to have extended root systems and dryland trees are known to develop a taproot that can access the deep water table (Do et al., 2008; Fan et al., 2017). This allows dryland evergreen trees to sustain carbon assimilation and transpiration throughout the dry season (Bowman and Prior, 2005; Nepstad et al., 1994; Oliveira et al., 2005; Whitley et al., 2017) as well as hydraulically redistributing water from moist to dry soil layers (Maeght et al., 2013; Wang et al., 2023). One of the current hypotheses in the development of deep roots states that dryland trees benefit from

occasional wet years by using the temporary increase in soil moisture as a window of opportunity to grow roots beyond the



otherwise 'dry gap' in the soil (Holmgren et al., 2013; Wang et al., 2023). The static representation of root architecture and root depth prevents ecosystem models such as LPJ-GUESS to simulate these dynamics, eventually underestimating the resilience of dryland trees to drought. Indeed, from our sensitivity tests it follows that root architecture can play a large role in determining the locations of soil water uptake. The current representation is based on a power law (Jackson et al., 1996) and

updating it to include dynamic rooting depth, deep taproots or an adaptive root biomass distribution scheme will allow our model to sustain evergreen dryland trees and dry-season transpiration, even for deep groundwater levels (Do et al., 2008; Maeght et al., 2013; Sakschewski et al., 2021).

**Conclusion**

In this work we presented an update to the LPJ-GUESS dynamic global vegetation model, in which we implented a process-
based representation of soil water movement by solving Richard's equation. This development is important for simulating dryland ecohydrology realistically, as soil water forms the reservoir for plans to take up water from. We showed that this update resulted in a generally better match with observations of carbon and water fluxes, although the improvements were overall small. We also showed that the updated model is more sensitive to soil texture. Furthermore, the new bottom boundary conditions opened up the model to simulate more ecosystem types, such as groundwater-dependent ecosystems. Tree cover
was overall lower in the new model, in favor of increased grass and shrub cover. We argued that including (1) a better representation of root architecture, including deep roots, and (2) an improved plant hydraulics scheme, could re-introduce simulated dryland trees in the model. Taken together, these developments will allow the LPJ-GUESS model to simulate dryland ecohydrology more realistically, enabling the scientific community to better understand and project the future of drylands under global change. The work presented in this paper forms the first milestone towards this goal.

**Code and data availability**

The RE-based branch of the LPJ-GUESS model can be downloaded from the following open access Zenodo archive: https://doi.org/10.5281/zenodo.15024130 (Verbruggen et al., 2025). The default version of the model can be downloaded from the following archive: https://zenodo.org/records/8065737 (Nord et al., 2021). Meteorological drivers and evaluation data can be obtained from the respective official repositories (Martens et al., 2017; Muñoz-Sabater et al., 2021; Tagesson et al., 2015;
Wieckowski et al., 2024).

**Author contribution**

WV designed the study, implemented the model development, performed the model runs and analysis, and wrote the manuscript. DW contributed to model development and revising the manuscript. FM and HV contributed to the soil texture



sensitivity studies and revising the manuscript. GS and SH supervised the entire process and contributed to revising the
manuscript.

**Financial support**

WV, SH and GS were supported by the HEIDA project (Reconcilling hydrological and ecological models to understand impacts of increasing drought and aridity; Geocenter Denmark, grant 5-2024) and the DRYTIP project (Drought-induced tipping points in ecosystem functioning; Villum Fonden, grant 37465). FM was funded by the FWO as a senior postdoc and
is thankful to this organisation for its financial support (FWO grant no. 1214723N). DW acknowledges funding from the European Union's Horizon Europe research and innovation programme under GreenFeedBack (grant no. 101056921).

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
