# Peer review of "Implementing a process-based representation of soil water movement in a second-generation dynamic vegetation model: application to dryland ecosystems (LPJ-GUESS-RE v1.0)"

_EGUsphere, 2025_

## Author Response (AR1)

Note: all replies to reviewers are formatted in *italics*. Quoted changes to the manuscript are formatted in ***bold italics***.

~ Reviewer 1 ~ ~ ~ ~ ~ ~ ~ ~ ~ ~ ~ ~ ~ ~ ~ ~ ~ ~ ~ ~ ~ ~ ~ ~ ~ ~ ~ ~ ~ ~ ~ ~ ~ ~ ~ ~ ~ ~ ~ ~ ~ ~ ~ ~ ~ ~ ~ ~ ~ ~ ~ ~ ~ ~ ~

Dear Authors,

I read, with great interest, your manuscript titled "Implementing a process-based representation of soil water movement in a second-generation dynamic vegetation model: application to dryland ecosystems (LPJ-GUESS-RE v1.0)".

The manuscript presents a new version of a dynamic vegetation model that introduces a more physically based representation of water transfer in soils using the Richards equation (RE). The new model is described in great detail, then applied at both a site and regional scale. The outputs are exhaustively discussed and compared to those obtained with default version of the model.

I do not have significant comments on the overall approach used here. However, there are opportunities for more clarification in certain aspects of the paper. Please find those detailed below. As such, I recommend publication of this paper after these minor comments will have been addressed.

General comments:

The main criticism I have for this paper is related to the use of constant soil properties for all layers of the soil profile. While this assumption can be acceptable for Arenosols like the one present at the studied site (since these soils are mostly sandy and have barely discernable horizons), it no longer holds for regional studies and even less for global ones. In fact, the Sudan-Sahel region studied here, hosts multiple soil types characterized by varying soil texture with depth (Acrisols, Vertisols, and even Ferralsols at its southern boundary just to name a few). The effects of texture on soil moisture have been shown by the sensitivity analysis conducted in this study, so having layer dependent soil properties will probably further influence the model outputs. While I understand that it is unrealistic to change this assumption now, I strongly recommend that future applications of the model take into account this depth-dependency, especially since: (i) the authors mentioned that this can be easily done (Line 174), and (ii) the ISRIC SoilGrids database (from which soil texture were obtained for this study) already provides values for six soil layers.

***REPLY:*** *We agree that using vertical soil would further increase the realism of the model, as well as having a potentially large impact on the simulated soil hydrology, vegetation cover and evaporation components. Including these variations would require a few changes in the model code, which we suggest to be included in the development priorities for the next version of this model. For this publication we aimed to keep both model versions as close to each other as possible, in order to compare the effects of the dynamics of process-based soil water movements on the model output. Including variations of soil texture with depth would bring our new model version one step further ahead of the default version, which would in turn make it harder to understand where exactly the differences in model output originate from. Therefore, we extended the discussion on the importance of including heterogenous soil texture columns in the discussion (see specific comment below). We also added*

*more information on the average soil texture, as well as the variations between the layers in the Africa SoilGrids (see the new section S3.3 in Supplementary Materials, the added references in the text, and our reply to the specific comment below).*

Specific comments:

Line 69: does the bedrock condition allow lateral runoff?

**REPLY:** *Yes it does, and the implementation of this is described in section 2.3.4. We did not make changes/clarifications for this.*

Line 138: Since soil layer thickness can vary in this new model version, is the evaporation layer defined as the two top soil layers or the top 20 cm? It is later stated (Line 232) that surface evaporation only occurs from the top layer (10 cm). Can you clarify the confusion between depth and number of layers when discussing evaporation domains?

**REPLY:** *In the default version of the model (LPJ-GUESS v4.1) evaporation can occur from the top two soil layers, corresponding to the top 20 cm. For our updated version of the model, evaporation can only occur from the top soil layer, as it is this layer which forms the interface between the soil and the atmosphere above it. By default, we set the thickness of this layer to 10 cm, but this can indeed vary if the user chooses a different size. In the new version presented in the manuscript, evaporation occurs strictly from the top layer, irrespective of its thickness. We clarified this by writing "**surface evaporation only occurs from the top layer (by default 10 cm)**" in section 2.3.4.*

Line 185: according to Ireson et al. (2023), the adaptative timestep ODE solver can use an implicit or explicit method. Which one was used here? This could be relevant to ensure numerical stability in the case of variable layer thickness.

**REPLY:** *We used the Runge-Kutta Cash-Karp adaptive timestepper (runge_kutta_cash_karp54 in ODEINT). This solver uses an explicit method. We now clarified this in section 2.3.2 as well as at the end of section 2.3.5: "**To integrate this system of ODEs we used the explicit Runge-Kutta Cash-Karp adaptive timestepper (runge_kutta_cash_karp54) from the odeint library in the boost C++ package (Ahnert and Mulansky, 2011; Cash and Karp, 1990)**." In addition, we also added a few lines in the discussion on further ways to improve computational efficiency: "**Using an implicit matrix-based approach to solving Richards equations would be a way forward towards increasing the computational efficiency, for example using a predictor-corrector method (Bonan, 2019). Providing the ODE solver with the Jacobian sparcity matrix has also been reported to improve computational efficiency if the ODE solver is capable of using this matrix as an input (Ireson et al., 2023).**"*

Line 288: Any information about at which depth this soil texture properties were measured?

**REPLY:** *This was described in the supplementary materials of Tagesson et al. (2015) as follows: "In February 2011, soil samples were taken at six different sites in the vicinity of the meteorological tower. At five of the sites three samples were taken in the first 20 cm. At the sixth site a soil profile up to 40 cm was made, and 4 samples were collected at 10, 20, 30 and 40 cm depth. Texture of the soil, and soil nitrogen*

*and carbon concentration were estimated." The authors did not describe in that paper how the texture was estimated, but T. Tagesson is a co-author of this revised manuscript, and he confirms that this was done by averaging the texture of each sample. We added this information on the sampling procedure in the revised manuscript, section 2.4:* **"Soil texture input (95.04% sand, 4.61% silt, 0.35% clay) was obtained from the average of local soil sample measurements at six sites in the vicinity of the flux tower, where five sites were sampled in the top 20 cm and a sixth site was sampled at 10, 20, 30 and 40 cm depth (Tagesson et al., 2015)."**

Lines 291-293: Can you provide some information about how much soil texture data varied between soil layers before averaging? This can help understand the potential importance/consequences of this assumption on the simulated soil hydrology. Also, a map of the averaged texture could be useful for interpretation of model performance. Finally, the same soil texture data were used for both the default and RE versions of the model, correct?

**REPLY:** *To account for these requests, we now added a new section (S3.3) to the supplementary materials, containing the map of averaged soil texture (Fig. S3), a map of the inter-layer coefficient of variation (Fig. S4), and a distribution plot for both figures, after selecting only those gridcells that were simulated (Fig. S5). From these figures it is clear that the inter-layer averages for most gridcells show a significant sand fraction (mean 0.546), followed by lower clay (mean 0.263) and silt (median 0.191) fractions. The inter-layer variation (CV) in sand and silt fraction is low (90% of gridcells had a CV<0.1) but variation in clay fraction is higher (90% of gridcells had a CV<0.23). We added a reference and a summary of this information to the Methods section (halfway Section 2.4), and we elaborate a bit more on this topic in the Discussion:* **"We showed that most gridcells in our regional simulation contain a relatively high sand fraction, with only a low vertical variation in sand content. However, the vertical variation in clay contents were higher and our soil texture sensitivity analysis showed that this may have an impact on soil water dynamics and the general outcome of our simulations. Accounting for vertical heterogeneity in soil properties requires a few significant, yet straight-forward, changes to the model code, which we suggest to prioritise for further model development."** *We confirm that the same soil texture data were used for both model versions.*

Lines 368-370 & Figure 3d: For the aquifer boundary condition, the groundwater table depth was considered to be equal to the soil profile depth (1.5 m), right? Given the big impact of this assumption on the soil moisture profile, I wonder how realistic this assumption is, especially for a dry site.

**REPLY:** *The goal of Figure 3 is to compare the different bottom boundary conditions in the new model version, and was not intended as a realistic case study. We added a clarification on this in section 3.1:* **"Note that the aquifer is only shown here at this shallow depth in order to compare the effect of the different bottom boundary conditions. It is not presented as a realistic representation of the Dahra fluxtower site conditions."**

Line 472 & Figure 9: To what depth do surface and root-zone soil moistures correspond?

**REPLY:** *Surface soil moisture corresponds to the upper 10 cm in the CCI/GLEAM dataproduct, while root-zone soil moisture is simulated over two layers (0-10 cm and 10-100 cm) for low vegetation types (e.g. grasslands) in the GLEAM model. We compared this against our model output by averaging over all layers between 0-100 cm. We clarified this in the main text (Section 2.5 and referred to Martens et al.*

(2017) for more information: *"**The GLEAM data product also includes assimilated surface (0-10 cm) soil moisture from the Climate Change Initiative (CCI) programme of the European Space Agency (ESA) (Dorigo et al., 2017; Gruber et al., 2017), as well as simulated root-zone soil moisture (0-100 cm for low vegetation types, e.g. grasslands) (Martens et al., 2017). Both were used for evaluating our soil moisture simulations, after averaging simulated soil moisture over the corresponding soil layers in our model.**"*

Lines 663-665: regarding the footprint of the rainy season that is still visible in measurements but not simulated by both versions of the model, the PTFs being partially responsible for this is a plausible explanation, but not the only one. Another one could be the role of preferential water flow through macropores and channels created by decaying roots. This kind of flow can bypass the soil matrix altogether and might be behind the footprint being still present in deeper layers. By definition, this preferential flow is not accounted for by the RE and would require introducing dual permeability flow in soils. This, of course, would add another level of complexity to an already quite complex model.

**REPLY:** *Thank you for this additional and interesting explanation. We mentioned soil water infiltration along roots in the original manuscript, which we elaborated a bit more on, adding a few references in the discussion: "**Nevertheless, other unaccounted processes can also play a large role in dryland soil moisture dynamics, such as hydraulic redistribution (Barron-Gafford et al., 2017; Bogie et al., 2018; Wang et al., 2023) and preferential water flow through along stems and roots, as well as macropores and channels created by decaying roots (Devitt and Smith, 2002; Xiao-Yan Li et al., 2009). These processes may also significantly contribute to the rainy season footprint that is visible in the data, but not in the model. We could account for these by using a double-porosity water retention curve (Cheng and Feng, 2023).**"*

Technical comments:

Line 12: "processes" instead of "processsess"

Line 14: add comma after "In this study".

Line 28 and elsewhere: add comma after "Recently". Consider this correction wherever it is relevant in the whole manuscript.

Line 35: cycling instead of "cyling"

Line 45: "This model was used in several of the earlier mentioned dryland studies".

Line 46: remove "-" after site.

S1.2 Eq.4: "wcont" instead of "wont"

Line 250: delete "a" in "using a an".

Line 261: sometimes the RE is referred to as "Richards equation" or "Richard's equation". Please use consistent naming.

Line 482: I think you meant to refer to Fig S7 here.

Line 584: "a" instead "an". Also, "ground water" should be written with no space.

Line 696: "error" instead of "erorr".

**REPLY:** *All technical comments are now addressed.*

~ Reviewer 2 ~~

Dear authors,

Thank you for writing and submitting this interesting manuscript documenting and testing the implementation of the Richards equation in LPJ-GUESS and comparing it to the bucket water hydrology used in the model before. My background is applied mathematics, so my review will naturally focus on the method section of the paper.

General comments

Overall, the description of the method and the results is clear, the method is mathematically solid and well justified, and the detailed analysis of the results provides interesting insights into soil hydrology modelling in dynamic vegetation models. Although the employed method is not new—it is taken from Ireson et al. (2023) with two slight adaptations (unequal layer thicknesses and a sink term)—the detailed analysis of the effects of using the Richards equation instead of a bucket water hydrology makes this manuscript a valuable resource for modelers in the community, thinking about implementing the Richards equation.

The method section is in general well written and mathematically clear. The description of the baseline model seems to be split in subsections 2.3.4 and subsection 2.2.2. Potentially, the generally well-organized method section could further benefit from moving all description of the baseline model (before the updates) to section 2.3.4. There are some equations and mathematical sentences that are likely erroneous, and should be corrected before publication. However, the errors can be easily corrected and the numerics set forth is still correct and uninfluenced by the errors.

**REPLY:** *We moved the initial reference of surface evaporation, transpiration and interception loss from section 2.3.4 to the baseline model description in section 2.2.2. The remainder of section 2.3.4 addresses the new model. For clarity, we decided to keep this description separated from the default model.*

The model i.e. the partial differential equation (PDE) written in equation (5), is mathematically equivalent to the standard Richards equation. It is formulated in terms of water potential and slightly changed compared to the classical formulation using the product rule on the time derivative of the soil water content to split this term into the derivative of water content w.r.t to water potential $C(\psi)$ and the time derivative of water potential. The only issue I find with the PDE is that the location of the sink term S

within the equation is likely wrong (see line specific comment). The equation is then discretized w.r.t to space, following Ireson et al. (2023). The approach of the reference is slightly generalized by considering layers with unequal thickness, which is mathematically well justified. The result is then a system of ordinary differential equations, written down in equation (6) and (7), for which different, physically meaningful, boundary conditions are applied. For the numerical integration an off-the-shelf numerical solver, that fits the application, is used.

**REPLY:** *We would like to thank the reviewer for pointing out the error in equation 5. We modified the equation accordingly, and would like to confirm that this was only a presentation issue with equation 5. The discretized version (Eq. 6) of this formula was correctly implemented in the original code and used for all simulations.*

Two simulations have been carried out, one at a fluxtower site at Dahra and another one for the whole Sudan-Sahel region with both, new and old, model versions. The results are comprehensively analyzed and the versions are compared to each other. In addition, both model results are confronted with reference data and their performance is evaluated. For the regional simulation, the model performance is clearly improved by using the Richards equation: For the surface soil moisture, the bias of the old version is almost gone in the new version and the correlation to reference data is significantly enhanced. Additionally, the transpiration bias is strongly reduced. For the Dahra site the use of the Richards equation does not improve performance, which could well be explained by site specific confounding factors, like missing livestock grazing, as you (the authors) suggest.

A very valuable insight for modelers of the community is that you demonstrate, how the sensitivity of the model outputs w.r.t to soil texture is increased, and more boundary conditions are possible with the new soil hydrology scheme. This shows how the Richards equation is more flexible and versatile w.r.t modelling different site conditions, compared to the bucket water scheme.

In the discussion, I suggest to additionally highlight as a clear improvement of the new model version, that the Richards equation results are more physically realistic, since spatial discontinuities in the solution are avoided.

**REPLY:** *We now added this highlight in the discussion: "**The resulting average soil moisture profile is more physically realistic, as it does not contain any of the sharp gradients that the default model version suffers from due to its layer grouping.**"*

Specific comments

1. Line 145 (equation (2)): It is unclear to me whether this represents total percolation that happens between the layers per day or the percolation rate. I think to make this concrete could be beneficial.

**REPLY:** *this represents the fraction of plant-available water that is transported between layers in one simulation timestep, ie. the total percolation between layers per day. We have added a line for clarification below Eq. 2: "**perc represents the fraction of plant-available water (wcont) that is transported between layers in one simulation timestep.**"*

2. Line 151: I understand that this is described in more detail in section 2.3.4. Without this further explanation I wasn't able to understand this sentence. My suggestion would be to provide the details of baseflow runoff and lateral flow runoff in the baseline model here and then only describe the changes made to these flow (if there are any) in section 2.3.4.

**REPLY:** *As suggested, we added some more details on the runoff components to the description of the baseline model (Sect. 2.2.2), as well as a reference to Haxeltine & Prentice (1996), where these calculations originate from: "**A fraction of plant-available water in the lower layers (50–150cm) further percolates out of the system as baseflow runoff, based on the percolation equation Eq. (2) divided by a factor 2. In addition, excess water in the lower layers dissipates out of the system as lateral flow runoff, similar to surface runoff (Haxeltine and Prentice, 1996).**" Section 2.3.4 now focuses on the updated calculations of these runoff components.*

3. Line 167 (equation (4)): The following question is likely only because of my lack of understanding of soil hydrological modelling. In the reference Campbell (1974) the formula $K = K_s(\theta/\theta_s)2b+2$ is used. The exponent $2b+3$ is only used when an interaction term is added. However, I don't see an interaction term being used here. The question is: What was the reason for choosing to use the exponent $2b+3$ instead of $2b+2$?

**REPLY:** *Campbell shows initially a formula that uses the $2b+b$ exponent, but then adds that models that use an interaction term give good results as well, leading to an exponent of $2b+3$. The abstract of Campbell mentions only the $2b+3$ exponent, and as far as we are aware, it seems most hydrologist these days have adapted the $2b+3$ exponent as the default (e.g. Bonan, 2019). This interaction term only refers to the empirical model that is used for calculating soil hydraulic conductivity, so it is not a process that needs to be added to our soil water dynamics model.*

4. Line 180 (equation (5)): The sink term S appears to be misplaced in this equation. Since the sink directly adds or removes water, it should be placed outside of the brackets of the spatial derivative, while still being multiplied with $1/C(\psi)$. It is not the spatial derivative of the sink that contributes to the volumetric water content rate of change $d\theta/dt$, but the sink term S directly. Thus, i suggest to correct equation (5) as:

$$d\psi/dt = 1/C(\psi) \, (d/dz \, (K(\psi) \, (d\psi/dz - 1) \,) - S) \quad (5)$$

In the spatially discretized equation (6) the sink term $Et_i/\Delta z_i$ is placed correctly, so the model code based on this should function correctly.

**REPLY:** *We would like to thank the reviewer for catching this formulation error. We modified Equation 5 accordingly. We also agree that this does not affect the results of the paper, as it is the (correct) discretized version of the equation (Eq. 6) that was used for all simulations.*

5. Line 184: The use of the term ordinary differential equation (ODE) is incorrect, I believe. Equation (5) is a partial differential equation (PDE), since it involves a spatial derivative w.r.t the variable x in addition to the temporal derivative. An ODE will only be achieved after the spatial discretization by the method of lines. That is equation (6) is then indeed an ODE. I suggest to use the term PDE for equation (5) and mention the "method of lines" to get from equation (5) to equation (6).

**REPLY:** *We corrected the terminology accordingly in Section 2.3.2 and mentioned the method of lines:* **"To obtain a numerical solution of this partial differential equation (Eq. (5)), we first used the method of lines to discretize this equation, resulting in Eq. (6)."**

6. Line 192 ("Eti is the implementation... "): When this equation is multiplied with $C(\psi_i)$ both sites of the equation represent the rate of change of soil water content theta for each layer. Then the term "$E_{ti}/\Delta z_i$" is precisely the sink term S in the corrected version of equation (5), but not "$E_{ti}$" itself. That is because equation (5) (when multiplied with $C(\psi)$) also has the rate of change (time derivative) of volumetric soil water content on both sides, but not the rate of change of absolute soil water content. So, S is the additionally rate of change of volumetric soil water content due to the sink.

**REPLY:** *Thank you for this clarification, we now added a note below Eq. (6) to clarify that we also divided $E_{ti}$ by $\Delta z_i$ to obtain change in volumetric water content:* **"Note that Eti was divided by Δzi to obtain the units of change in volumetric water content, so in Eq. (6) it is Eti/Δzi which corresponds to S in Eq. (5)."**

7. Line 240: Why is the R_drain flux calculated at the end of the day and not handled analogously to the other fluxes Es, Et and Win, that is passed to the sub-daily RE integrator?

**REPLY:** *In our current implementation, the calculation of R_drain is more consistent with the calculation of the other fluxes, such as evaporation, surface runoff and infiltrating water. This is because all of these are updated with a daily timestep in the current implementation. If we would include R_drain inside the RE integrator this would be updated with a subdaily timestep. Furthermore, the values of R_drain are overall relatively low, compared to for example surface runoff. We do not think that calculating R_drain with a subdaily timestep would create a large difference. Baseflow runoff (R_base) is however calculated with a subdaily timestep, as this is part of the bottom boundary conditions and is also calculated using RE.*

8. Line 262 ("To ensure water mass balance..."): As I understand the technique of Ireson (2023) the use of the additional equations $dQ_1/dt = q_1$ and in the ODE system $dQ_N/dt = q_N$ is not to ensure water mass balance closure, but to calculate the cumulative boundary fluxes, and thus to evaluate the deviation from water mass balance closure, i.e. to evaluate equation (13).

**REPLY:** *Agreed, we reformulated/corrected the beginning of this line:* **"To allow for calculating water mass balance..."**

9. Line 266 (equation (14)): To make this equation more clear, I suggest to either use an integral over the timespan [0,tnow] or to use a sum from i=0 to i=K, over $q_j(t_i)$ where K is the number of timesteps, i.e. sum over discrete times $t_1,...,t_K$. The way this equation is written now, the sum seems to be taken over a continuous interval (uncountable many terms) which is at least not a standard mathematical thing to do.

**REPLY:** *Agreed, we now write it as the sum over discrete timesteps $[t\_1..t\_T]$ instead in Eq. (14).*

10. Line 463 ("due to a large spatial variability..."): Why does the large spatial variability in the correlation lead to a lower average correlation

**REPLY:** *This is because, although many gridcells show a positive correlation, there are also with a pronounced anticorrelation (Fig. 8c). When taking the average over all gridcells, these negative correlation values reduce the average correlation. We rewrote this line to make this more explicit: "**The correlations between either model version and the GLEAM transpiration time-series over 1980–2022 were overall positive, but low on average (R=0.25 for both models) because there are also several gridcells that showed an anticorrelation, e.g. in the western parts of the region (Figure 8c).**"*

11. Line 634 ("updating the soil water dynamics in LPJ-GUESS resulted ..."): Based on my interpretation of Figure 4, Figure 5, Table 1, Figure S5 for the Dahra site simulation and Figure 7 for the regional simulation compared to GLEAM, there is indeed an improved performance for the regional simulation when using the Richards equation, but not for the Dahra site simulation.

**REPLY:** *We would argue that the Dahra site simulation does show a small improvement, for example when evaluating the rainy season peak ET, as well as soil moisture content for the upper layers. However, we added a line to provide some more nuance: "**However, the improvements were only small overall, and especially so for the site level simulations.**"*

An additional advantage I see for using RE is that, for both simulations (regional and Dahra) the soil moisture profile "looks" much better using the RE, that is sharp gradients without physical meaning (which I would call artefacts of the numerical method) are avoided (see e.g. Figure S7).

**REPLY:** *Agreed, we now mentioned this further in the discussion: "**The resulting average soil moisture profile is more physically realistic, as it does not contain any of the sharp gradients that the default model version suffers from due to its layer grouping.**"*

12. Line 643 ("Including these boundary conditions..."): Maybe add: "may therefore further improve the overall model performance [for the regional simulation]"? For the Dahra site this seems not to be the case, as I understand the explanation of line 621-623.

**REPLY:** *If adding a ground water table is the only change to the model, then we agree that his improvement will be only for the regional simulation. Therefore, we added "**regional**" to this line (first paragraph of Discussion). However, adding a deep ground water table may also improve the simulation for the Dahra site once we also include a representation of deep roots. Future updates to the rooting architecture are already discussed at the end of the Discussion, so we did not mention this here.*

13. Line 696 ("This also resulted..."): I would be interested in understanding this matrix-based approach better. Is it an implicit method (like backward Euler or trapezoidal method)? Using an implicit method (based on matrices) could indeed increase computational efficiency since no restriction of the length of the timestep is needed for stability and one may be able to avoid sub-daily time stepping all together. However, using an implicit method would also mean that a nonlinear equation system would needed to be solved each timestep.

**REPLY:** *We added the following line to the Discussion: "**Using an implicit matrix-based approach to solving Richards equations would be a way forward towards increasing the computational efficiency, for example using a predictor-corrector method (Bonan, 2019). Providing the ODE solver with the Jacobian sparcity matrix has also been reported to improve computational efficiency if the ODE solver is capable of using this matrix as an input (Ireson et al., 2023).**"*

14. Line 718 ("Indeed, from our sensitivity..."):  I was unable to locate this sensitivity analysis in the paper. Does this sentence refer to Figure 14?

**REPLY:** *This indeed refers to Fig. 14 but is elaborated in the next sentence in the discussion. We mean that including an adaptive root distribution scheme, as well as deeper roots and a taproot may give the vegetation in our model a better access to the soil moisture where it is available, such as near the groundwater table. We rephrased the sentence to make it more general, without referring to a specific sensitivity test:* **"Indeed, our results suggest that..."** *(last paragraph of Discussion).*

Technical corrections

1. Line 148 ("see Sup. Mat. S1.1"): A typographic comment: Could you write this out as in line 131 for consistency?

**REPLY:** *Addressed.*

2. Line 234 ("The removal of water... "): This sentence could be read to imply that the water infiltration Win is not included in the ODE solver runtime. However, paragraph 2.3.5 clearly states that Win is passed to the ODE solver analogously to Et and Es. My suggestion is to add "Win" to the list of things implemented inside the ODE solver routine, if this is the case.

**REPLY:** *Win is not defined yet at this point, so we do not think that this change is necessary. It is clarified in Section 2.3.5 that these three variables are passed on to the ODE solver.*

3. Line 525 (Figure 10): For easier readability and more consistency it would be beneficial to use dotted lines for default values, as in Figure 2. (Or change Figure 2 respectively.)

**REPLY:** *Addressed, we changed the linetypes in Fig. 10.*

4. Line 528 ("meteoroloigical"): This is a typo.

**REPLY:** *Addressed.*

5. Line 618 ("the the"): This is a typo.

**REPLY:** *Addressed.*

6. Line 653 ("down to -1.4 m"): Is the minus sign in front of 1.4m a typo?

**REPLY:** *This was just to clarify that we are looking downward. However, it is inconsistent with the axis labels in the figures, so we removed the minus sign. For consistency, we also removed the minus signs from the y-axes in Figures 3 and S7.*